# FHL1 promotes chikungunya and o'nyong-nyong virus infection and pathogenesis with implications for alphavirus vaccine design

Wern Hann Ng[1,2,3,12], Xiang Liu[1,2,3,12], Zheng L. Ling[4,5], Camilla N. O. Santos[6], Lucas S. Magalhães [6], Andrew J. Kueh[7,8], Marco J. Herold [7,8], Adam Taylor[1,2,3], Joseph R. Freitas[1,2,3], Sandra Koit [9], Sainan Wang [9], Andrew R. Lloyd [10], Mauro M. Teixeira[11], Andres Merits[9], Roque P. Almeida[6], Nicholas J. C. King [1,4,5] & Suresh Mahalingam [1,2,3] ✉

Arthritogenic alphaviruses are positive-strand RNA viruses that cause debilitating musculoskeletal diseases affecting millions worldwide. A recent discovery identified the four-and-a-half-LIM domain protein 1 splice variant A (FHL1A) as a crucial host factor interacting with the hypervariable domain (HVD) of chikungunya virus (CHIKV) nonstructural protein 3 (nsP3). Here, we show that acute and chronic chikungunya disease in humans correlates with elevated levels of FHL1. We generated FHL1$^{-/-}$ mice, which when infected with CHIKV or o'nyong-nyong virus (ONNV) displayed reduced arthritis and myositis, fewer immune infiltrates, and reduced proinflammatory cytokine/chemokine outputs, compared to infected wild-type (WT) mice. Interestingly, disease signs were comparable in FHL1$^{-/-}$ and WT mice infected with arthritogenic alphaviruses Ross River virus (RRV) or Mayaro virus (MAYV). This aligns with pull-down assay data, which showed the ability of CHIKV and ONNV nsP3 to interact with FHL1, while RRV and MAYV nsP3s did not. We engineered a CHIKV mutant unable to bind FHL1 (CHIKV-ΔFHL1), which was avirulent in vivo. Following inoculation with CHIKV-ΔFHL1, mice were protected from disease upon challenge with CHIKV and ONNV, and viraemia was significantly reduced in RRV- and MAYV-challenged mice. Targeting FHL1-binding as an approach to vaccine design could lead to breakthroughs in mitigating alphaviral disease.

Arthritogenic alphaviruses are a large group of positive-strand RNA viruses known to cause acute and chronic musculoskeletal disease[1,2]. Notable examples, including chikungunya virus (CHIKV), o'nyong-nyong virus (ONNV), Ross River virus (RRV) and Mayaro virus (MAYV), cause recurrent episodic outbreaks and have shown the capacity to cause widespread epidemics[3–5]. Despite their debilitating impact on human health and potential for global emergence, no licensed therapies or vaccines currently exist[6,7].

Alphaviruses encode four replicase proteins named nonstructural proteins 1-4 (nsP1-nsP4). NsP1, nsP2 and nsP4 form the membrane-bound core of viral RNA replicase. NsP3 is indispensable for viral replication and is known to interact with numerous host proteins[8]. Host factors such as Ras-GAP SH3 domain binding proteins (G3BPs), amphiphysin[9], Y-box-binding protein 1 (YBX1)[10], heat shock proteins[10] and NAP1L1/4[11] interact with alphavirus nsP3[12–15]. The C-terminal hypervariable domain (HVD) of nsP3 has been shown to be

important for host protein interaction, however, the role of nsP3 HVD in alphavirus pathogenesis remains unclear[11,16–18].

Four-and-a-half LIM domain protein 1 (FHL1, also known as SLIM1) is a unique member of the LIM protein family that was recently discovered to play a key role in CHIKV and ONNV RNA replication[19]. FHL1 is involved in multiple cellular processes. Mutations in the FHL1 gene cause many distinct heart and musculoskeletal diseases, which have been categorized into FHL1-related myopathies or FHL1opathies[20]. Notable examples include reducing body myopathy, Emery-Dreifuss muscular disorder (EDMD), myofibrillar myopathy (MFM), X-linked recessive hypoparathyroidism, and hypertrophic cardiomyopathy. Many of these diseases are progressive, and left untreated, have various implications for the quality of life of the affected individual, eventually resulting in death[21–25]. Three major FHL1 splice variants – FHL1A, FHL1B and FHL1C – are found in humans. FHL1A is most commonly found in skeletal and cardiac muscle and fibroblasts, whereas FHL1B and FHL1C are found mostly in muscle, brain and testes[26–29]. FHL1A was identified as the splice variant that plays a role in CHIKV and ONNV infections. It was reported that the deletion of FHL1 led to inhibition of CHIKV and ONNV infection, but did not affect other alphaviruses, such as RRV and MAYV, or flaviviruses. Notably, FHL1 was found to interact directly with CHIKV nsP3 HVD. The study further showed the cells from a patient with EDMD, deficient in FHL1, were resistant to CHIKV infection. Furthermore, based on limited in vivo studies using 9-day-old genetically engineered mice that lacked FHL1 (FHL1$^{-/y}$), Meertens et al. could not detect CHIKV replication in these mice. Histologically, there were no signs of necrosis and reduced cellular infiltrates compared to WT mice[19]. Interestingly, Lukash et al. recently reported that the FHL1-nsP3 HVD interaction may not be a prerequisite of viral replication, as CHIKV was still able to replicate in cells deficient for FHL1[30]. Our results agree with Lukash et al. where we were able to detect replication of CHIKV, albeit attenuated in CHIKV-infected FHL1$^{-/-}$ mice. The role of FHL1 in the development of alphavirus diseases has not been fully characterized.

In our study, we demonstrate for the first time that FHL1 levels are increased in patients with acute and chronic chikungunya disease. We also generated FHL1 knockout (FHL1$^{-/-}$) mice on the C57BL/6J background and used them to examine the role of FHL1 in arthritogenic alphavirus disease. Knockout of FHL1A without affecting FHL1B/C is technically difficult, as the three isoforms have heavily overlapping exons. In mice infected with either CHIKV or ONNV, joint swelling was diminished in FHL1$^{-/-}$ compared to wild-type (WT) mice. Additionally, data obtained from histopathological analysis, immunoassay and mass cytometry (CyTOF) demonstrated reduced lesions, lower expression of proinflammatory cytokines and reduced cellular infiltrates in infected FHL1$^{-/-}$ mice compared to WT mice. In contrast, FHL1$^{-/-}$ mice infected with either RRV or MAYV did not show reduced disease symptoms. We carried out a pull-down assay, which demonstrated a distinct interaction between FHL1 and the nsP3s of CHIKV and ONNV, but no such interaction was observed with the nsP3s of RRV and MAYV. Additionally, we created a CHIKV mutant (CHIKV-ΔFHL1) unable to interact with FHL1 (confirmed by pull-down assay) to test its pathogenicity and efficacy as a universal vaccine candidate against arthritogenic alphaviruses. CHIKV-ΔFHL1 infection was highly attenuated in Vero cells, showing significantly reduced replication kinetics compared to CHIKV-WT. Disease signs were significantly reduced in CHIKV-ΔFHL1 inoculated mice compared to CHIKV-WT infected, with CHIKV-ΔFHL1 inoculation offering protection from disease development upon subsequent challenge with CHIKV, ONNV and, to a degree, RRV and MAYV.

## Results

### FHL1 was upregulated during CHIKV infection
Viral infection leads to a range of changes in host factors to facilitate viral replication[16]. To determine whether levels of FHL1 are increased during CHIKV infection, FHL1 protein levels in serum samples from chikungunya patients were quantified. Serum samples collected from chikungunya patients with either acute or chronic disease, thirty-nine and seventeen respectively, were analyzed by ELISA; samples from healthy individuals were used as controls (Supplementary Table 1 and 2). FHL1 protein levels were significantly higher in the serum of both acute and chronic chikungunya patients compared to the healthy controls (Fig. 1a). Similar observations were made in a mouse model of acute CHIKV disease. Mice were infected with CHIKV-WT or mock-infected subcutaneously in the right footpad. At 1-, 3- and 5-days post-infection (dpi), blood was collected via tail bleeding, and the ipsilateral quadriceps were harvested at 7 dpi. The quadriceps were processed for wholemount immunofluorescence staining of FHL1 (the antibody was produced by GenScript Biotech based on Q13642-1, UniProt). We observed an increased FHL1 antibody signal in the quadriceps of infected mice compared to mock-infected mice (Fig. 1b), recapitulating our observations in CHIKV patients. To investigate if the elevated levels of FHL1 at the sites of inflammation is detectable on a transcription level, total RNA was extracted from quadriceps and FHL1 mRNA expression analyzed by qRT-PCR. A significant increase in FHL1 mRNA was observed in the ipsilateral quadriceps of CHIKV-infected mice at 7 dpi (Fig. 1c); a time which coincides with the second peak of foot swelling (Fig. 2a). Moreover, CHIKV-infected mice exhibited increased serum levels of FHL1 across all time points (1, 3 and 5 dpi) compared to mock-infected mice (Fig. 1d). These results indicate that expression of FHL1 is elevated as a result of CHIKV infection.

### The severity of CHIKV disease was less pronounced in FHL1$^{-/-}$ mice
To define the role of FHL1 in CHIKV disease pathogenesis, WT and FHL1$^{-/-}$ mice were inoculated subcutaneously in the footpad with $10^4$ PFU of CHIKV or mock-infected with PBS. CHIKV-infected FHL1$^{-/-}$ mice showed significantly reduced foot swelling compared to infected WT mice (Fig. 2a). Reduced CHIKV titers were observed in the serum of FHL1$^{-/-}$ mice compared to that of WT mice at 1-, 3-, and 5 dpi (Fig. 2b). Furthermore, in both ipsilateral and contralateral quadriceps and ankle joints, CHIKV-infected FHL1$^{-/-}$ mice had significantly reduced virus titers compared to WT mice at 3 and 7 dpi (Fig. 2c–f).

Hematoxylin and eosin (H&E) staining of the quadriceps and ankles was used to enumerate cell infiltrates in these tissues. In mock-infected WT and FHL1$^{-/-}$ mice, comparable cellular infiltrates were observed in the quadriceps (Fig. 3a, b) and ankle joints (Fig. 3g, h). Compared to CHIKV-infected WT mice, infected FHL1$^{-/-}$ mice showed reduced cellular infiltrates in the ipsilateral quadriceps at 3- and 7 dpi (Fig. 3c–f). Interestingly, no significant differences were found at 3 dpi in the ipsilateral ankles of CHIKV-infected FHL1$^{-/-}$ and WT mice (Fig. 3i, j). However, at 7 dpi, infected WT mice showed significantly higher cellular infiltrates in the ipsilateral ankles than infected FHL1$^{-/-}$ mice (Fig. 3k, l).

Additionally, the ankles of CHIKV-infected mice collected at 3 and 7 dpi were stained with safranin-O to assess differences in articular cartilage thickness. For mock-infected mice, the articular cartilage thickness was comparable between WT and FHL1$^{-/-}$ mice (Supplementary Fig. 2a, b). Ipsilateral ankles of CHIKV-infected FHL1$^{-/-}$ and WT mice also exhibited similar articular cartilage thickness at 3 dpi (Supplementary Fig. 2c, d). In contrast, a notable reduction in articular cartilage thickness was observed in the ipsilateral ankle of infected WT mice compared to infected FHL1$^{-/-}$ mice at 7 dpi (Supplementary Fig. 2e, f). Taken together, these results demonstrate FHL1 is necessary for the development of acute CHIKV disease in mice.

### CHIKV-infected FHL1$^{-/-}$ mice exhibit reduced levels of proinflammatory cytokines and limited immune cell infiltration
Next, we examined the effects of FHL1 knock-out on the cytokine profile and immune response following CHIKV infection in mice. The

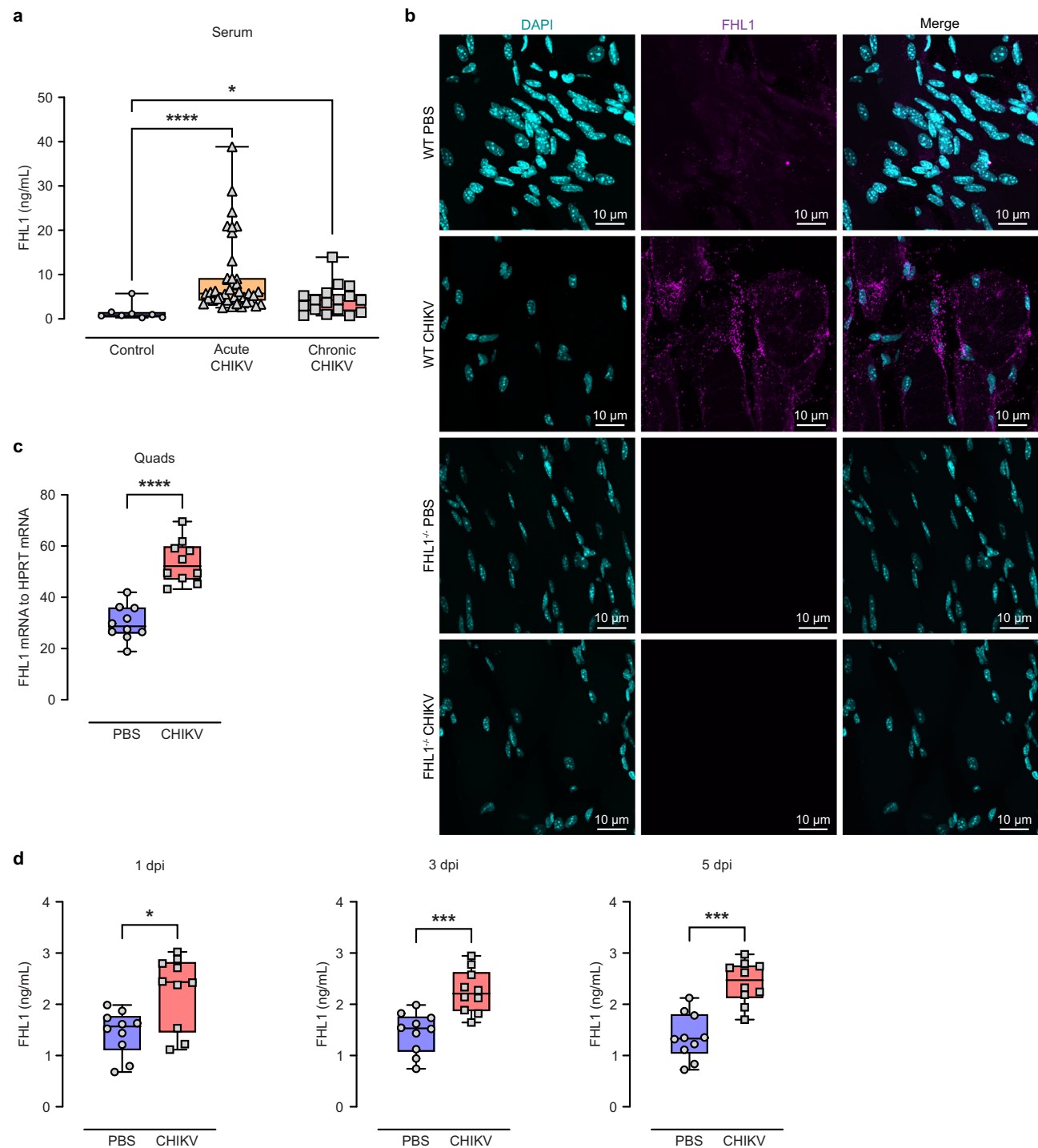

**Fig. 1 | FHL1 levels are increased in CHIKV infected humans and mice.** The levels of FHL1 in the serum from acute and chronic CHIKV disease patients were determined by ELISA (**a**). Data are presented as box and whisker ± SD with the mean indicated by a line across the box, maximum to minimum points. Dots represent individual participant (healthy individuals (control): $n = 8$; acute CHIKV patients: $n = 39$; chronic CHIKV patients: $n = 17$ (*$P < 0.05$; ****$P < 0.0001$; one-way ANOVA with the Kruskal–Wallis posttest). Mice were infected with CHIKV at $10^4$ PFU or mock-infected with PBS. Serum was collected at 1, 3 and 5 dpi, and the ipsilateral quadriceps were harvested at 7 dpi. The quadriceps were processed for

immunofluorescence staining (**b**) and qRT–PCR analysis of FHL1 mRNA (**c**). The confocal immunofluorescence microscopy images shown are representative of $n = 5$ mice per group; the data are representative of two independent experiments. The serum was processed for ELISA analysis of FHL1 (**d**). Dots represent individual animals ($n = 10$). Data are representative of two independent experiments. Data are presented as box and whisker ± SD with the mean indicated by a line across the box, maximum to minimum points (*$P < 0.05$; ***$P < 0.001$; ****$P < 0.0001$; Mann–Whitney test) (**c**, **d**). Source data are provided as a Source Data file.

levels of proinflammatory cytokines and chemokines, including G-CSF, GM-CSF, IFN-γ, IL-1β, IL-6, IL-12p70, IL-17A, CCL-2, CCL-3, CCL-4, CCL-5, CXCL1 and TNF-α in the ipsilateral quadriceps were significantly lower in FHL1$^{-/-}$ mice compared to WT mice both at 3 and 7 dpi (Fig. 4). This

was also observed for Eotaxin, IL-1α, IL-2, IL-3, and IL-12p40 levels at 7 dpi, but not at 3 dpi. Interestingly, the anti-inflammatory IL-10 and the allergy-associated cytokines, IL-9 and IL-13, were also significantly lower in FHL1$^{-/-}$ than WT mice at both time points (Fig. 4a).

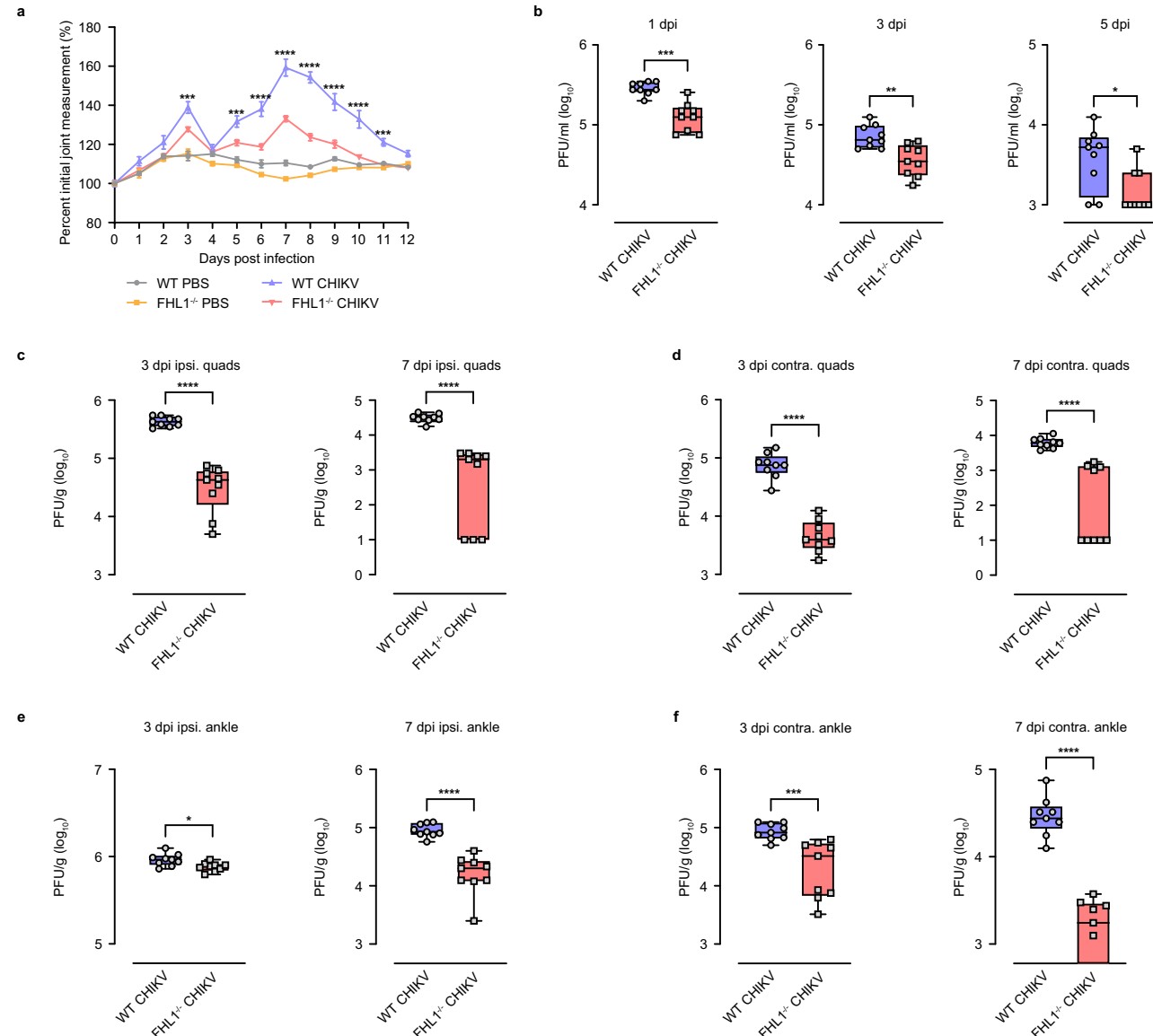

**Fig. 2 | FHL1 facilitates CHIKV disease development and viral replication in mice.** WT and FHL1[−/−] mice were infected with CHIKV at 10[4] PFU or mock-infected with PBS. Disease was monitored daily and assessed by measuring the height and width of the perimetatarsal area of the ipsilateral hind foot (**a**). The data shown are representative of two independent experiments; $n = 10$ mice per group (***$P < 0.001$; ****$P < 0.0001$; two-way ANOVA with the Bonferroni posttest). Serum was collected at 1, 3 and 5 dpi and processed for plaque assays (**b**). Dots represent individual animals ($n = 9$). The data shown are representative of two independent experiments. Data are presented as box and whisker ± SD with the mean indicated by a line across the box, maximum to minimum points (*$P < 0.05$; **$P < 0.01$; ***$P < 0.001$; Mann−Whitney test). The contralateral and ipsilateral quadriceps and ankles were harvested at 3 and 7 dpi. Viral titers were determined by a plaque assay (**c**−**f**). Dots represent individual animals ($n = 9$). The data shown are representative of two independent experiments. Data are presented as box and whisker ± SD with the mean indicated by a line across the box, maximum to minimum points (*$P < 0.05$; ***$P < 0.001$; ****$P < 0.0001$; Mann−Whitney test). Source data are provided as a Source Data file.

The ipsilateral quadriceps, popliteal draining lymph nodes (DLNs) and spleens of CHIKV-infected FHL1[−/−] and WT mice were collected and examined with Mass Cytometry and fluorescent flow cytometry. There were no significant differences in major immune cell populations observed between WT and FHL1[−/−] mice at 0 or 3 dpi in the ipsilateral quadriceps (Fig. 4b) or DLNs (Supplementary Fig. 3a), while a reduction in some populations was detected in spleens of FHL1[−/−] mice at 3 dpi (Supplementary Fig. 3b). In contrast, at 7 dpi, macrophages, monocytes, natural killer (NK) cells, CD4[+] and CD8[+] T cells were all significantly reduced in the ipsilateral quadriceps (Fig. 4b), DLNs (Supplementary Fig. 3a) and spleen (Supplementary Fig. 3b) of FHL1[−/−] mice, compared to WT mice.

Mass cytometry data of DLNs and spleens were processed by Uniform Manifold Approximation and Projection (UMAP) for dimension reduction and visualization. Clusters of major cell types were identified on the UMAP plots based on the heatmaps of the expression levels of fundamental cell markers, such as CD3, CD4, CD8, CD11b, CD19, etc. Position shifts on the UMAP plots represent phenotype changes of the cells (Supplementary Fig. 4a, f). Cell markers that principally contributed to CHIKV-induced shifts included CCR7, Ly6A/E (Sca-1) and Ly6C in various cell types. In the DLNs, these surface antigens were found to be increased on T and B lymphocytes, as well as CD11b[+] myeloid cells in both CHIKV-infected FHL1[−/−] and WT mice (Supplementary Fig. 4b−e). However, while cells in the infected WT mice tended to maintain relatively high levels of CCR7, Ly6A/E and Ly6C expression over time, levels were significantly lower in the infected FHL1[−/−] mice, with this difference greater at 7 dpi (Supplementary Fig. 4b−e). The differential expression between CHIKV-

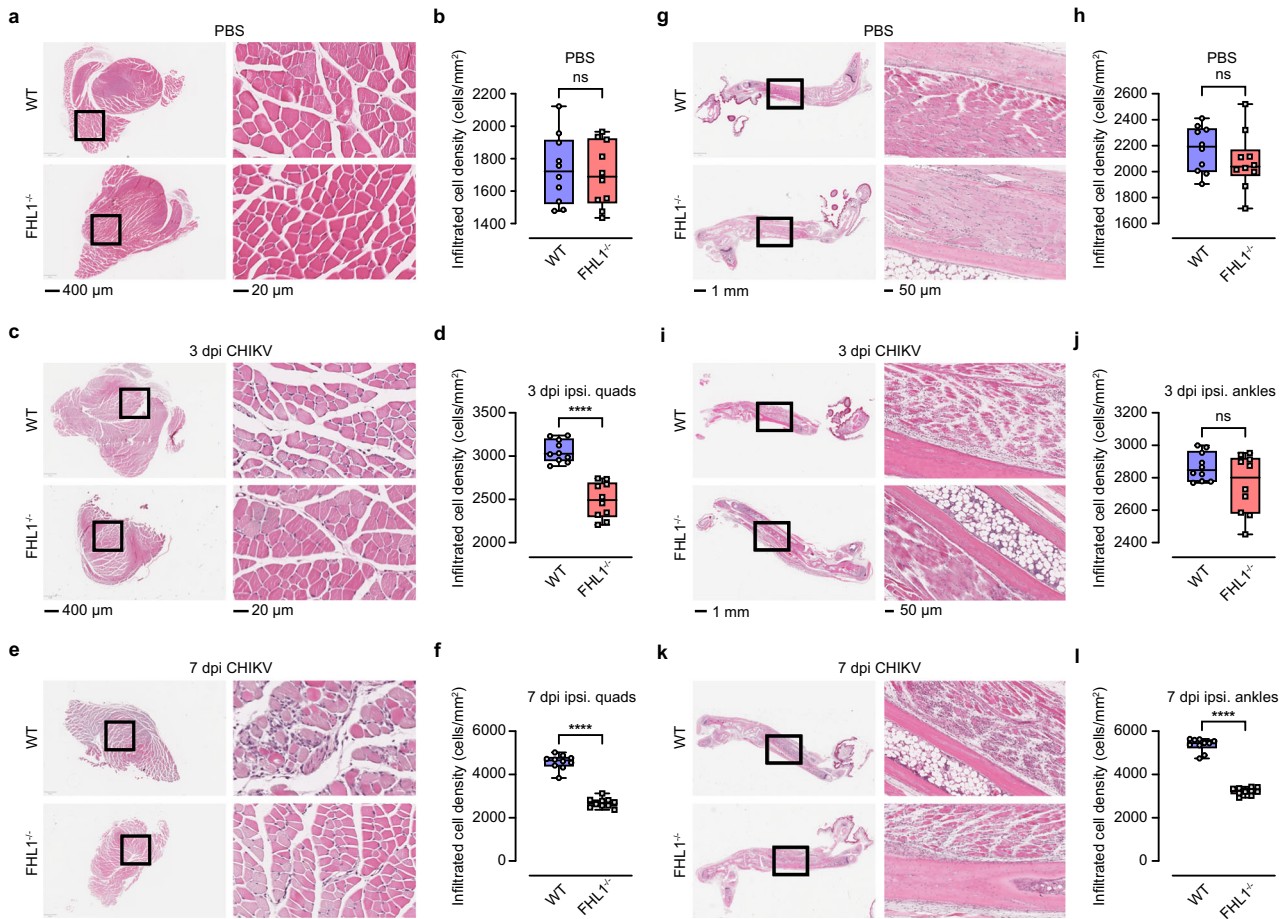

**Fig. 3 | FHL1 facilitates cell infiltration in the quadriceps and ankles of mice with CHIKV disease.** WT and FHL1$^{-/-}$ mice were infected with CHIKV at $10^4$ PFU or mock-infected with PBS. The ipsilateral quadriceps were collected at 3 and 7 dpi and processed for H&E staining (**a**, **c**, **e**). The microscopy images shown are representative of $n = 10$ mice per group. The numbers of infiltrated cells were analyzed using ImageScope [Min Nuclear Size ($\mu m^2$) = 20; Max Nuclear Size ($\mu m^2$) = 200; Min Roundness = 0.4; Min Compactness = 0.4; Min Elongation = 0.2]. Dots represent individual animals ($n = 10$); the data shown are representative of two independent experiments (**b**, **d**, **f**). Data are presented as box and whisker ± SD with the mean indicated by a line across the box, maximum to minimum points (ns,

nonsignificant; ****$P < 0.0001$; Mann−Whitney test). The ipsilateral ankles were collected at 3 and 7 dpi and processed for H&E staining (**g**, **i**, **k**). The microscopy images shown are representative of $n = 10$ mice per group. The numbers of infiltrated cells were analyzed using ImageScope [Min Nuclear Size ($\mu m^2$) = 20; Max Nuclear Size ($\mu m^2$) = 200; Min Roundness = 0.4; Min Compactness = 0.4; Min Elongation = 0.2]. Dots represent individual animals ($n = 10$); the data shown are representative of two independent experiments (**h**, **j**, **l**). Data are presented as box and whisker ± SD with the mean indicated by a line across the box, maximum to minimum points (ns, nonsignificant; ****$P < 0.0001$; Mann−Whitney test). Source data are provided as a Source Data file.

infected FHL1$^{-/-}$ and WT mice was also observed in splenocytes at 7 dpi, although interestingly, in contrast to the DLNs, Ly6C expression was significantly higher in B cells and CD11b$^+$ myeloid cells at 3dpi (Supplementary Fig. 4g–j). The profile of these cell surface antigens is in line with the lower disease symptoms observed in CHIKV infected FHL1$^{-/-}$ mice. Furthermore, the higher levels of Ly6C in the infected FHL1$^{-/-}$ mice suggests a potential novel role for Ly6C and monocytes in the early stage of CHIKV disease pathogenesis.

We further investigated whether recruited leukocytes in infected mouse muscle contributed to elevated FHL1 expression in this tissue during disease using immunofluorescence staining. Several CD4, Ly6C/G and CD11b cell clusters in the myofiber interstitial space expressing high levels of FHL1 were observed, indicating that myeloid cells such as monocytes and macrophages were important, albeit not majorly, for secreted FHL1 during peak CHIKV disease (Supplementary Fig. 5).

## Impact of FHL1 on other arthritogenic alphavirus infections in vivo
From the observations above, we hypothesized that FHL1 might also play a significant role in infection and disease of other clinically important arthritogenic alphaviruses such as RRV, MAYV and ONNV.

To determine if FHL1 expression is increased during RRV infection, we collected serum and plasma from patients with confirmed RRV disease and from healthy individuals (Supplementary Table 3) and quantified FHL1 protein levels by ELISA. Similar to the results observed in samples from chikungunya patients, RRV disease patient serum and plasma samples exhibited significantly higher levels of FHL1 than samples from healthy controls (Fig. 5a, b). Interestingly, however, in a mouse model of acute RRV disease, we did not observe any significant differences in the disease score between RRV-infected WT and FHL1$^{-/-}$ mice (Fig. 5c, d). Consistent with this observation, viral titers in serum samples collected at 1-, 3- and 5-dpi (Fig. 5e), and those in quadriceps and ankle samples at 3- and 7-dpi (Fig. 5f, g), were comparable between RRV-infected WT and FHL1$^{-/-}$ mice. Similar observations were made using a mouse model of MAYV disease; disease severity and virus titers in the serum, quadriceps and ankles were all comparable in MAYV-infected WT and FHL1$^{-/-}$ mice (Fig. 5h-k). In contrast, upon ONNV infection, we observed a significant reduction in foot swelling in FHL1$^{-/-}$ mice compared to WT mice. The reduction in foot swelling in FHL1$^{-/-}$ mice was noted as early as 3 dpi and significant differences were observed up to 10 dpi (Fig. 5l). Consistent with this observation, FHL1$^{-/-}$ mice exhibited reduced ONNV titers in the serum, ipsilateral

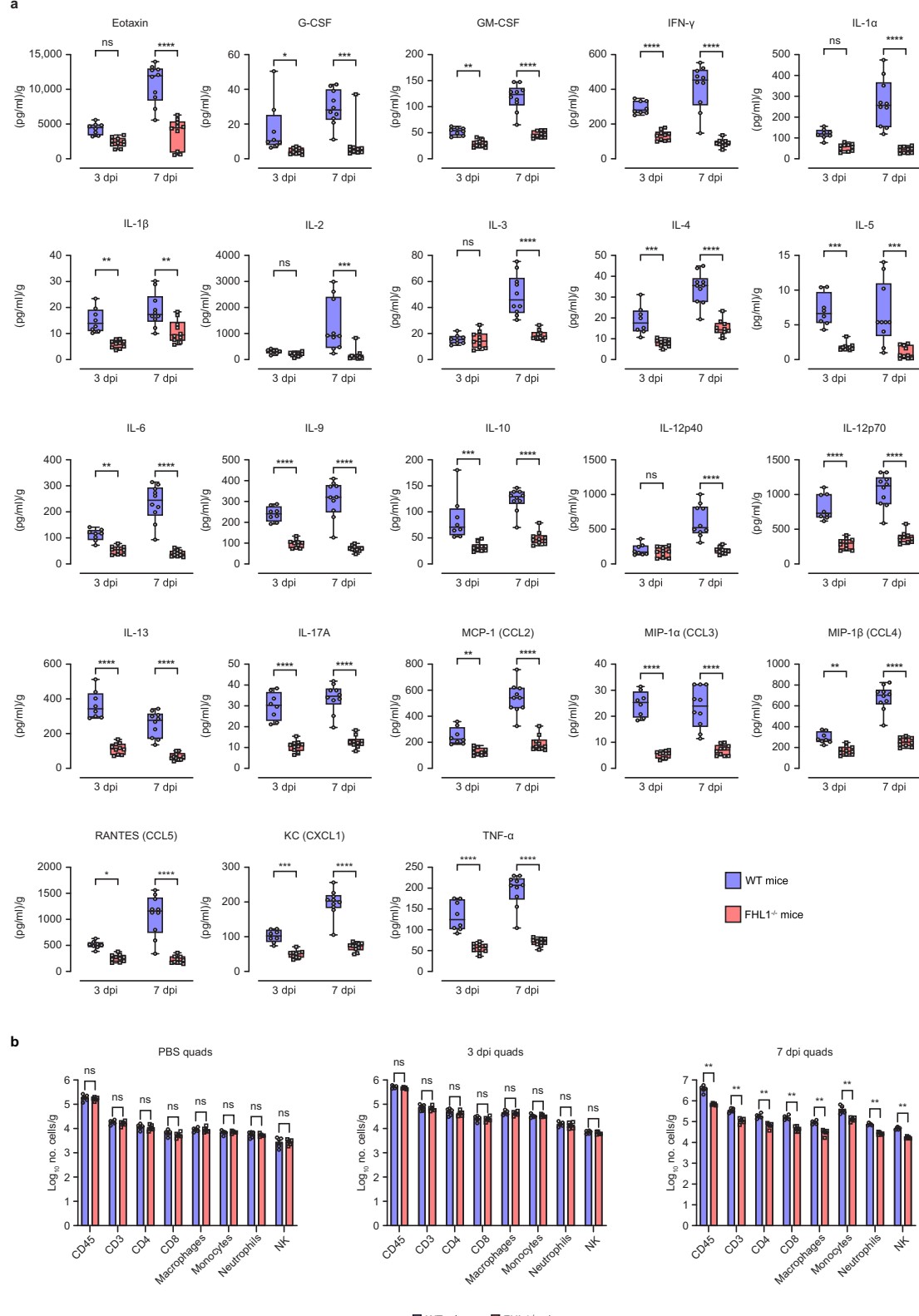

**Fig. 4 | Knock-out of FHL1 reduces levels of proinflammatory cytokines and chemokines in CHIKV-infected mice.** WT and FHL1$^{-/-}$ mice were infected with CHIKV at 10$^4$ PFU. The ipsilateral quadriceps were collected at 3 and 7 dpi and processed for a cytokine/chemokine multiplex assay (a). Dots represent individual animals ($n = 8$, WT mice; $n = 10$, FHL1$^{-/-}$ mice); the data shown are representative of two independent experiments. Data are presented as box and whisker ± SD with the mean indicated by a line across the box, maximum to minimum points (ns,

nonsignificant; *$P < 0.05$; **$P < 0.01$; ***$P < 0.001$ ****$P < 0.0001$; two-way ANOVA with the Holm−Sidak posttest). WT and FHL1$^{-/-}$ mice were infected with 10$^4$ PFU CHIKV or mock-infected with PBS. The ipsilateral quadriceps were harvested at 3 and 7 dpi and processed for flow cytometry (b). Dots represent individual animals ($n = 5$); the data shown are representative of two independent experiments. Data are presented as the mean ± SEM from (ns nonsignificant; **$P < 0.01$; Mann−Whitney test). Source data are provided as a Source Data file.

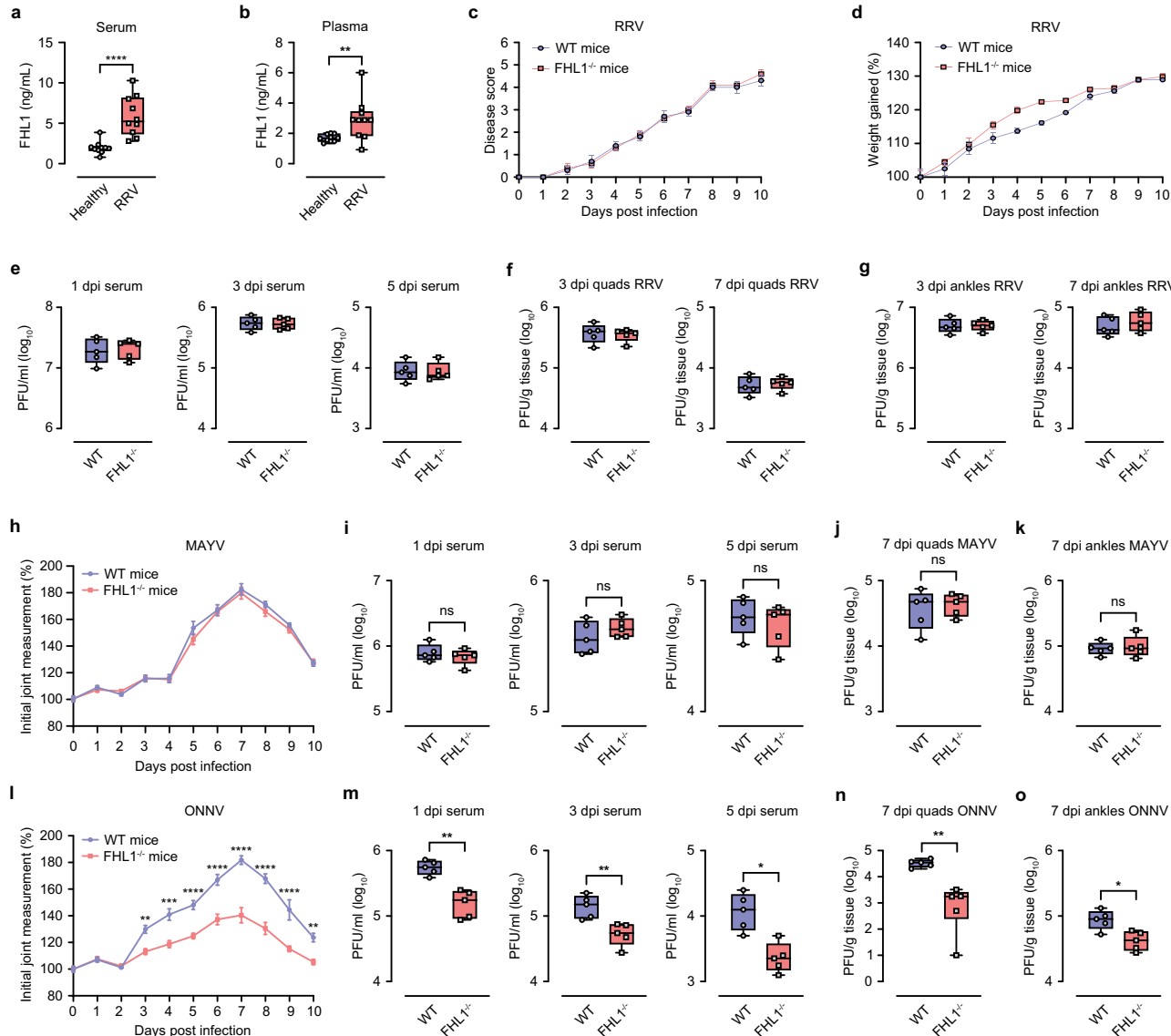

**Fig. 5 | Impact of FHL1 on RRV, MAYV and ONNV infection.** The levels of FHL1 in the serum and plasma from RRV disease patients were determined by ELISA (**a**, **b**). Dots represent individual participant (*n* = 10) (\*\**P* < 0.01; \*\*\*\**P* < 0.0001; Mann-Whitney test). WT and FHL1$^{-/-}$ mice were infected with RRV at 10$^4$ PFU. Disease score and weight were monitored daily (**c**, **d**), (*n* = 5; two-way ANOVA with the Bonferroni posttest). Mouse serum was collected at 1, 3 and 5 dpi and processed for plaque assays (**e**). Mouse quadriceps and ankles were harvested at 3 and 7 dpi. Viral titers were determined by a plaque assay (**f**, **g**). For **e**–**g**, dots represent individual animals; *n* = 5 mice per group. WT and FHL1$^{-/-}$ mice were infected with MAYV at 10$^4$ PFU. Mouse disease was monitored daily (**h**). (*n* = 5; two-way ANOVA with the Bonferroni posttest). Mouse serum was collected at 1, 3 and 5 dpi and processed for plaque assays (**i**). Mouse quadriceps and ankles were harvested at 3 and 7 dpi. Viral titers were determined by a plaque assay (**j**, **k**). For **i**–**k**, dots represent individual animals; *n* = 5 mice per group (ns, nonsignificant). WT and FHL1$^{-/-}$ mice were infected with ONNV at 10$^4$ PFU. Mouse disease was monitored daily (**l**). (\*\**P* < 0.01; \*\*\**P* < 0.001; \*\*\*\**P* < 0.0001; *n* = 5; two-way ANOVA with the Bonferroni posttest). Mouse serum was collected at 1, 3 and 5 dpi and processed for plaque assays (**m**). Mouse quadriceps and ankles were harvested at 3 and 7 dpi. Viral titers were determined by a plaque assay (**n**–**o**). For panel m to o, dots represent individual animals; *n* = 5 mice per group (\**P* < 0.05; \*\**P* < 0.01; Mann–Whitney test). Data are presented as box and whisker ± SD with the mean indicated by a line across the box, maximum to minimum points for **a**, **b**, **e**–**g**, **i**–**k** and **m**–**o** (Mann–Whitney test). Source data are provided as a Source Data file.

quadriceps, and ankles compared to WT mice at all time points analyzed (Fig. 5m–o).

Additionally, the replication kinetics of CHIKV, RRV, ONNV, and MAYV in primary murine fibroblasts derived from WT and FHL1$^{-/-}$ mice was analyzed using multistep growth curves. The obtained data was consistent with our in vivo data. Replication of CHIKV and ONNV was reduced in FHL1$^{-/-}$ fibroblasts compared to WT fibroblasts, whereas RRV and MAYV replication kinetics were similar in both cell types (Fig. 6a–d). Next, we fused HVD of nsP3 of RRV, MAYV, CHIKV and ONNV to EGFP, expressed these fusion proteins in U2OS cells and performed a pull-down assay experiment. It was found that while nsP3s

of all four viruses interacted with G3BP1, an essential host co-factor for replication of arthritic alphaviruses[31], only nsP3s of CHIKV and ONNV were able to interact with FHL1 (Fig. 6e). Combined, our results indicate that FHL1 is essential for CHIKV and ONNV infection and disease development, but not for RRV or MAYV.

## CHIKV mutant defective for interaction with FHL1 shows attenuated replication in Vero cells and offers protection from subsequent alphavirus challenge in mice

In previous studies, deletions in nsP3 HVD (R$_{1686}$-P$_{1795}$ and A$_{1796}$-D$_{1839}$; here and other places residues numbered from the beginning of

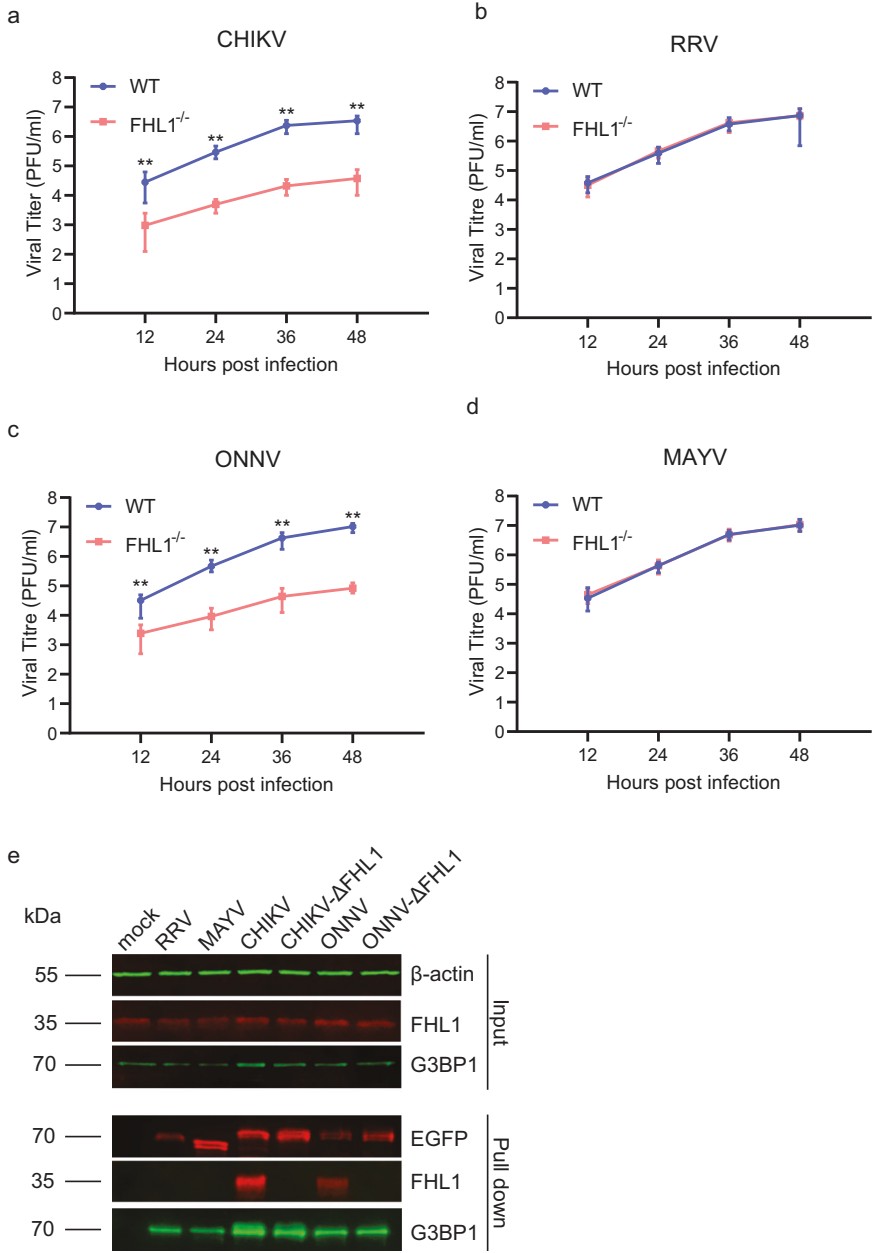

**Fig. 6 | FHL1 is essential for the replication of CHIKV and ONNV.** Multi-step growth curves of CHIKV (**a**), RRV (**b**), ONNV (**c**) and MAYV (**d**). Primary murine fibroblasts derived from WT or FHL1$^{-/-}$ mice were infected at an MOI of 0.01. Data are presented as the mean ± SD from two experiments each performed in five technical replicates (**\*\****P* < 0.05; Mann–Whitney test). EGFP pull-down experiment (**e**). U2OS cells were transfected with plasmids expressing EGFP fused with HVD of nsP3 of RRV, MAYV, CHIKV, CHIKV-ΔFHL1, ONNV or ONNV-ΔFHL1. At 24 h post transfection cells were harvested, lysed, and cellular proteins interacting with HVD of nsP3 of used viruses were pulled down using EGFP-binding magnetic beads. Proteins in input and in pull-down fraction were analyzed using SDS-PAGE and immunoblotting. Input: detection of G3PB1 and FHL1 in cell lysates before pull-down; antibodies against β-actin were used to detect loading control. Pull-down: proteins pulled down with EGFP-binding magnetic beads were detected with antibodies against EGFP, G3BP1 and FHL1. Images from one out of two reproducible experiments are shown. Source data are provided as a Source Data file.

nonstructural polyprotein) led to attenuation of Semliki Forest virus (SFV) virulence in mice[32]. In 2014, Hallengard et al. showed CHIKV-3del5, a CHIKV mutant with a large deletion in nsP3 by substituting E$^{1656}$ to G$^{1717}$ with a linker (amino acid sequence AYRAAAG), exhibited reduced replicative capacity and elicited high immunogenic protection in mice against CHIKV challenge, making it a promising CHIKV vaccine candidate (phase III clinical trial NCT04546724 completed)[33]. The critical amino acids mediating the interaction between nsP3 of CHIKV and FHL1 were reported to be V$_{1745}$(or P$_{1756}$) to P$_{1787}$[19,30], which are located within HVD of nsP3 (aa residues 1653–1863). We therefore hypothesized that mutations in nsP3 of CHIKV, preventing its

interaction with FHL1, should lead to reduced replication and attenuation in disease and that such an attenuated variant could be explored as a potential live-attenuated vaccine candidate against CHIKV infection.

First, mutations substituting six amino acids (F1762G, F1763G, T1780R, A1781S, Q1785G and A1786T) were introduced into a construct expressing EGFP fused with HVD of nsP3 of CHIKV (CHIKV-ΔFHL1); a matching set of mutations was also introduced into a construct expressing EGFP fused with HVD of nsP3 of ONNV (ONNV-ΔFHL1). Pull-down assays performed with these constructs revealed that the introduced mutations completely abolished interaction of HVDs of

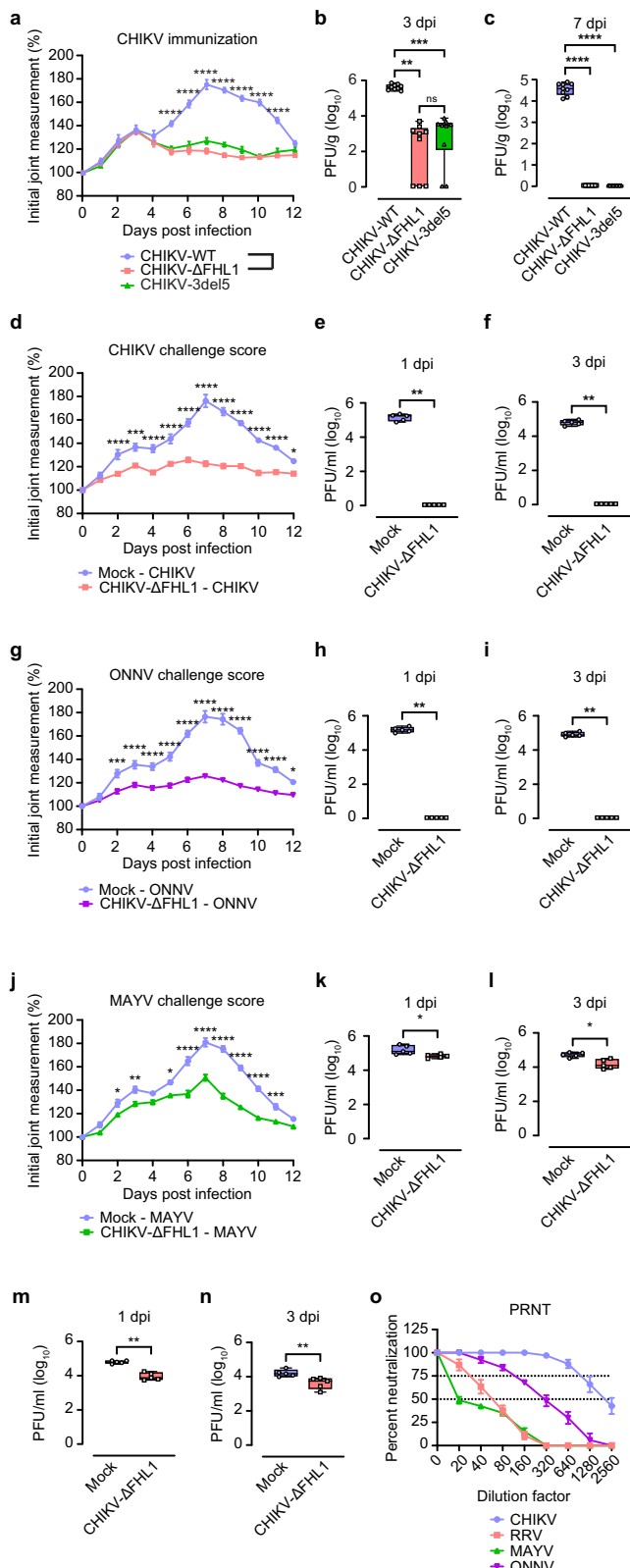

**Fig. 7 | CHIKV-ΔFHL1 is attenuated and protects vaccinated mice against infection of different alphaviruses.** Four-week-old WT mice were inoculated with $10^4$ PFU CHIKV-ΔFHL1, CHIKV-3del5 or CHIKV-WT. Disease were monitored daily (**a**). The data are representative of two independent experiments (****$P < 0.0001$; $n = 9$; two-way ANOVA with the Bonferroni posttest). The ipsilateral quadriceps were harvested at 3 and 7 dpi. Viral titers were determined by a plaque assay (**b**, **c**). Dots represent individual animals ($n = 9$). The data shown are representative of two independent experiments. Data are presented as box and whisker ± SD with the mean indicated by a line across the box, maximum to minimum points (ns, non-significant; **$P < 0.01$; ***$P < 0.001$; ****$P < 0.0001$; Mann–Whitney test). Four-week-old WT mice were vaccinated with $10^4$ PFU CHIKV-ΔFHL1 or mock-vaccinated with PBS on the ventral/lateral side of the right foot. The mice were challenged with $10^4$ PFU CHIKV (**d**), ONNV (**g**), MAYV (**j**) or RRV on day 16 post-vaccination. Disease scores (except for RRV-infected mice) were monitored and assessed daily by measuring the footpad. The data shown are representative of two independent experiments ($n = 5$ mice per group). All values represent the mean ± SEM (*$P < 0.05$; **$P < 0.01$; ***$P < 0.001$; ****$P < 0.0001$; two-way ANOVA with the Bonferroni post hoc test). Mouse serum was collected at 1 and 3 dpi. The viral titers in the serum of CHIKV- (**e**, **f**), ONNV- (**h**, **i**), MAYV- (**k**, **l**), and RRV-challenged (**m**, **n**) mice were determined by a plaque assay. Dots represent individual animals ($n = 5$); the data shown are representative of two independent experiments. Data are presented as box and whisker ± SD with the mean indicated by a line across the box, maximum to minimum points (*$P < 0.05$; **$P < 0.01$; Mann–Whitney test). The serum (collected at 14 days post-vaccination) neutralizing antibody levels were determined by a PRNT assay (**o**). Data are presented as mean ± SEM. Dotted lines signify the $PRNT_{50}$ and $PRNT_{75}$. The data shown are representative of two independent experiments ($n = 5$ mice per group). Source data are provided as a Source Data file.

CHIKV and ONNV with FHL1 while interactions with G3BP1 were not affected (Fig. 6e). The differences in the amino acid sequences of nsP3 HVD between CHIKV-WT, CHIKV-ΔFHL1 and CHIKV-3del5 are shown in Supplementary Fig. 6a. Similar to CHIKV-3del5, CHIKV-ΔFHL1 replicated to significantly lower titers in Vero cells compared to WT CHIKV, indicating a high degree of attenuated replication (Supplementary Fig. 6b). To assess the disease-causing capacity of CHIKV-ΔFHL1, WT

mice were infected subcutaneously in the right footpad with $10^4$ PFU CHIKV-WT, CHIKV-ΔFHL1 or CHIKV-3del5, and monitored daily for CHIKV-induced footpad swelling. At 3 dpi, the first peak of foot swelling was observed in all the three groups of infected mice. However, beginning from 5 dpi until 11 dpi, a significant reduction in foot swelling was observed in CHIKV-3del5 and CHIKV-ΔFHL1-infected mice, compared to CHIKV-WT-infected mice (Fig. 7a). Furthermore, CHIKV-ΔFHL1 and CHIKV-3del5 replicated to a significantly lower titer in quadriceps muscle at day 3 post infection, compared to CHIKV-WT (Fig. 7b). At day 7 post infection, the titer of CHIKV-ΔFHL1 and CHIKV-3del5 were below the detection limit (Fig. 7c). These results suggest that, similar to CHIKV-3del5, CHIKV-ΔFHL1 is highly attenuated in vivo and may therefore have potential as a vaccine candidate.

Next, we sought to determine the protective effects of vaccination with CHIKV-ΔFHL1. WT mice were inoculated subcutaneously with $10^4$ PFU of CHIKV-ΔFHL1 in the right footpad or were mock-vaccinated. At 16 days post vaccination, the mice were subsequently challenged with $10^4$ PFU of CHIKV, RRV, MAYV or ONNV. CHIKV-, MAYV- and ONNV-challenged mice were monitored for footpad swelling daily. Disease scores were not analyzed for RRV-challenged mice as this model requires mice to be 21 days of age upon infection to observe clinical disease signs[34–36]. Our data demonstrates that CHIKV-ΔFHL1 inoculated mice did not exhibit any CHIKV- or ONNV- induced foot swelling upon challenge (Fig. 7d, g) and had significantly reduced disease signs during MAYV challenge (Fig. 7j). Additionally, there was no detectable viremia in CHIKV-ΔFHL1 inoculated mice challenged with CHIKV (Fig. 7e, f), or ONNV at 1- and 3-days post-challenge (Fig. 7h, i). In CHIKV-ΔFHL1 inoculated mice challenged with MAYV (Fig. 7k, l) or RRV (Fig. 7m, n), while viremia was observed, virus titers were significantly reduced (by $0.5 - 0.8 \log_{10}$), compared to control mice challenged with these viruses. These results indicate that CHIKV-ΔFHL1 vaccination can protect mice from arthritogenic alphavirus-induced disease and viremia.

The humoral immune response, particularly neutralizing antibodies, are an important component of the host defense mechanism providing protection against alphaviruses[37–40]. Thus, the neutralizing capacity of antibodies induced by vaccination is indicative of the efficacy of the vaccine. Hence, we examined the neutralizing capability of

serum from CHIKV-ΔFHL1-inoculated mice collected at 14 days post-vaccination using plaque reduction neutralization test (PRNT) against CHIKV, RRV, MAYV or ONNV. A 50% PRNT (PRNT$_{50}$) value for CHIKV at ~1:2500 dilution was detected, whereas the PRNT$_{50}$ was at 1:160 to 1:320 for ONNV, at 1:40 to 1:80 for RRV, and <1:20 for MAYV (Fig. 7o). Thus, PRNT$_{50}$ values correlated with observed protection against different alphaviruses. These results suggest that the neutralizing antibodies induced by CHIKV-ΔFHL1 are not only able to protect against CHIKV infection, but also offer cross-protection against other alphaviruses, including closely related ONNV as well as more distantly related RRV and MAYV.

## Discussion

Previous studies have identified several host factors, including FHL1, G3BPs, glycosaminoglycans, T-cell immunoglobulin, mucin 1 and MXRA8 facilitate entry and replication of alphaviruses[12–15,19]. However, despite extensive efforts, many gaps remain in our current understanding of the role(s) that host factors play in alphavirus infection and pathogenesis.

In the current study, we sought to characterize the role of FHL1 in the pathogenesis of alphavirus diseases. We found that FHL1 levels were significantly higher in the serum of acute/chronic CHIKV disease and RRV disease, which, to the best of our knowledge, represents the first report showing the elevation of FHL1 upon alphavirus infection. Presumably, the high levels of FHL1 are derived from the breakdown of infected muscle. The level of FHL1 in the patient serum was higher in the acute phase of the disease than in the chronic phase. These observations suggest FHL1 may have potential to be used as a marker CHIKV disease diagnosis.

Previously, our study identified that pentraxin 3 (PTX3), a host factor that activates complement pathways and facilitates pathogen recognition by macrophages and DCs, was highly induced in CHIKV and RRV disease patients[40,41]. With this present study showing upregulation of FHL1, both of our studies suggest a novel strategy by which alphaviruses are likely to facilitate infection by upregulating certain host factors which are critical for virus replication. The underlying mechanisms and the repercussions for disease pathology warrant further investigation.

Our knowledge regarding the pathogenesis of chronic CHIKV disease is still limited. Persistence of viral replication in the joint tissues has been hypothesized to contribute to chronic CHIKV disease. It is therefore tempting to speculate that the ongoing high levels of FHL1 in chronic CHIKV infection may facilitate the persistence of infection which leads to chronic disease development.

Notably, FHL1 has been reported to increase muscle fiber size and oxidative slow fiber type expression, which are both associated with muscle development and strength[42]. Thus, the increased FHL1 levels in the serum of CHIKV and RRV disease patients, apart from facilitating virus replication, may also reflect tissue recovery. FHL1 therefore may have a dual role in alphavirus disease pathogenesis, which warrants further investigation. Whether increased levels of FHL1 are a host-driven response to muscle damage, a response to soluble factor(s) from infiltrating leukocytes, or are directed by the virus, as seen in flavivirus infection to immunopathological effect[43–45], remains to be investigated.

A previous study identified FHL1A as a key host factor required for optimum CHIKV and ONNV infections in vitro[19] while another study demonstrated it to be important but not critical[30]. While these findings are not necessarily mutually exclusive, they leave open the question of the importance of FHL1 for CHIKV infection and pathogenesis. Here, we comprehensively assessed the role of FHL1 in CHIKV infection and disease by employing FHL1-deficient mice in our well-established immunocompetent adult mouse model of infection that mimics CHIKV human disease (i.e. arthritis and myositis)[35]. FHL1 deficiency did not eliminate the ability of CHIKV to replicate in vitro or in vivo. However,

it was associated with significant reduction in CHIKV replication and foot swelling in mice. Inflammatory monocytes, macrophages and T cell infiltrates promote acute arthritogenic alphavirus disease[34,46]. Indeed, infected mice without FHL1 presented with lower cellular infiltrates and milder inflammation in the quadriceps and ankles than infected WT mice. The reduction in disease signs and cellular infiltrates were likely due to the inability of CHIKV to replicate efficiently which, in turn, is likely a consequence of the lack of interaction between the HVD of nsP3 and FHL1 required for optimal replication[19,30]. We have previously reported that CHIKV can replicate in bone and elicit bone loss by upsetting the RANKL/OPG ratio[47,48]. In the present study, we observed reduced cartilage damage in mice lacking FHL1, compared to WT mice. Concurrently, reduced levels of several proinflammatory mediators, particularly IL-6 and CCL-2 which were previously shown to facilitate an increase in the RANKL/OPG ratio leading to increased osteoclastogenic activities, were observed.

To investigate whether FHL1 is equally important in other alphavirus diseases, we evaluated the role of FHL1 in the inflammation induced by ONNV, MAYV or RRV, three arthritogenic alphaviruses that are related to CHIKV. We observed a reduction in ONNV-induced foot swelling in FHL1$^{-/-}$ mice. The differences observed in RRV and MAYV-induced disease, characterized by arbitrary disease score and foot swelling, respectively, were comparable between FHL1$^{-/-}$ and WT mice. These observations are consistent with a previous study showing that FHL1 is required only for optimal CHIKV and ONNV infection in vitro. FHL1 is known to interact with CHIKV nsP3 HVD. Phylogenetic analysis based on alphavirus nsP3 has shown several clades within the alphavirus genus, with CHIKV more closely related to ONNV and more distantly related to RRV and MAYV[49]. Consistent with this, our analysis confirmed that only nsP3 of CHIKV and ONNV were able to interact with FHL1 supporting our hypothesis that FHL1:nsP3 interaction is the key factor in the dependency of CHIKV and ONNV, but not RRV or MAYV on FHL1 for optimal infection. Our findings emphasize the complexity of alphavirus-host interactions. The requirement for FHL1 by CHIKV and ONNV for efficient replication may represent a distinct evolutionary path, compared to RRV and MAYV.

We detected significantly lower levels of proinflammatory cytokines and infiltrating leukocytes in the quadriceps of infected FHL1$^{-/-}$ mice than WT mice at 3 and 7 dpi. This is consistent with the reduced disease signs observed in these mice in our current study, as well as previous studies where higher levels of proinflammatory cytokines, neutrophils, macrophages, and monocytes are associated with more severe CHIKV diseases[34,50–53]. To assess the immune response induced by alphavirus infection in FHL1-deficient mice comprehensively, we employed the CyTOF Helios, which allowed detailed characterization of immune cell populations in the spleen and DLNs. Analysis of DLNs and splenocytes from CHIKV-infected mice using mass cytometry and dimension-reduction clustering revealed significant cell phenotype shifts due to upregulation of Ly6A/E, Ly6C and CCR7 on T and B lymphocytes, as well as cells from the myeloid lineage. Whether this is a result of cell activation and phenotype change, or due to differential migration of cell subsets, remains to be determined. Ly6A/E and Ly6C have previously been shown to be upregulated on T and B lymphocytes under inflammatory conditions[54–56], but to our knowledge, this has not been described in an alphavirus mouse model. Furthermore, the results showed that while these markers were induced by CHIKV infection in FHL1$^{-/-}$ mice, their expression levels were not sustained and returned to a lower state by 7 dpi. In contrast, expression levels of Ly6A/E, Ly6C and CCR7 on leukocytes in infected WT mice in almost all cases remained significantly higher than FHL1$^{-/-}$ mice at this point. CCR7 is primarily known for its role in homing of leukocytes to lymphoid organs in response to CCL19 and CCL21 signals[57], which may explain the abundance of this receptor in the DLNs and spleen in CHIKV-infected mice. Taken together, we consistently observed a lower level of CHIKV-induced inflammatory response in the DLNs,

spleens and muscles in the FHL1$^{-/-}$ mice than WT mice, which correlates with the reduced viral load in knockout mice.

Although there are a few promising vaccine candidates for CHIKV[33,58,59], there are as yet no licensed vaccines against alphavirus infection[6,7]. A number of live-attenuated vaccine candidates have been developed based on deletions or mutations of viral proteins[33,59,60]. Hallengard et al. developed a promising CHIKV live-attenuated vaccine candidate (CHIKV-3del5) by deleting a large fragment in the HVD of nsP3[33]. Vaccination of mice with CHIKV-3del5 induced a robust antibody response and provided protection from CHIKV challenge. Here, we demonstrated that CHIKV-ΔFHL1, a mutant virus containing mutations in nsP3 that prevent its interactions with FHL1, has the potential to act as a live-attenuated vaccine. In vitro growth kinetics of CHIKV-ΔFHL1 showed that if replicated to a significantly lower titer in Vero cells, compared to CHIKV-WT, confirming the attenuation of the virus. Similar to CHIKV-3del5, CHIKV-ΔFHL1-infected mice did not exhibit major CHIKV-induced foot swelling, highlighting the safety of this potential vaccine candidate. Vaccinated mice challenged with CHIKV-WT at 16 days post vaccination showed no signs of footpad swelling and did not develop noticeable viremia in the serum from days 1 to 3 post-challenge. Vaccination with CHIKV-ΔFHL1 also resulted in strong cross-protection against other alphaviruses. Protection was strongest against closely related ONNV but was significant against more distantly related MAYV and RRV that, similar to CHIKV, belong to the Semliki Forest antigenic complex of alphaviruses. It remains unknown whether vaccination with CHIKV-ΔFHL1 offers protection against alphaviruses belonging to other antigenic complexes. However, the broad spectrum of protection demonstrated in this study clearly represents an added advantage, making CHIKV-ΔFHL1 a potentially valuable universal vaccine candidate against multiple alphaviruses. Extensive research has been conducted on cross-reactive antibodies in relation to various viral infections. In developing vaccines for flaviviruses, influenza viruses, HIV and COVID-19, one of the objectives is to generate cross-reactive and broadly neutralizing antibody responses that can provide protection against infections of other virus members from the same family[61–65]. Some approved vaccines, such as smallpox vaccine and HPV vaccine, have been reported to cross protect against other closely related virus strains[66]. However, in some cases, such as dengue virus infections, pre-existing, cross-reactive immune responses may lead to exacerbated disease severity through antibody-dependent enhancement (ADE)[67]. For alphavirus infection, E1-specific monoclonal antibodies (mAb) isolated from Eastern equine encephalitis virus (EEEV) patients were shown to have broad cross-reactivity against encephalitic and arthritogenic alphaviruses[68]. CHK-265, a mouse CHIKV mAbs, has shown protection against RRV infection in mice[69]. To the best of our knowledge, our study is the first to demonstrate the effectiveness of a vaccine candidate protecting against CHIKV and the related alphaviruses ONNV, MAYV, and RRV. This is likely due to the cross-reactive antibodies stimulated by our vaccine candidate, as evidenced by the PRNT assay. Importantly, our data suggests a novel strategy of live-attenuated vaccine development by modifying the ability of a viral protein to interact with host factors. The potential of CHIKV-ΔFHL1 to be developed as an alphavirus vaccine candidate warrants further investigation.

The choice of appropriate control groups plays a critical role in ensuring the validity of experimental outcomes. Ideally, utilizing litter mate controls would provide a more robust comparison and control as it would minimize potential confounding factors such as microbiome-related variables. However, it is important to point out that FHL1$^{-/-}$ mice are not commercially available, which necessitated their in-house breeding following their generation. To minimize the chances of genetic drift, we crossbred FHL1$^{-/-}$ mice with WT counterparts sourced from the Australian Animal Resources Centre (ARC). Importantly, the ARC-sourced WT mice concurrently served as the control group for

our study. We have also validated our group attributes using flow cytometry and have shown that there was no significant difference observed in the populations of the major immune cells between the uninfected FHL1$^{-/-}$ and WT mice (Fig. 4b, Supplementary Fig. 3). Although we acknowledge that the absence of littermate controls represents a limitation in our study, we emphasize that the careful consideration of breeding methods, as well as flow cytometry validation, lends substantial support to the comparability of our experimental groups. These measures collectively contribute to the validity and reliability of the conclusions drawn from this study.

Taken together, our study results provide a detailed characterization of the role of FHL1 in alphavirus pathogenesis. Reduced disease signs and foot swelling were observed in CHIKV and ONNV infections in the absence of FHL1. In contrast, the absence of FHL1 did not affect disease progression in the context of RRV or MAYV infection. Vaccination with a CHIKV nsP3 mutant that is unable to interact with FHL1 can effectively protect mice against CHIKV, ONNV, RRV and MAYV challenge. While further studies are warranted to define the endogenous role of FHL1, our study also highlights the potential to target FHL1 as a therapeutic strategy for reduction of disease caused by CHIKV infection.

## Methods

### Ethics statement

All animal experiments were approved by the Animal Ethics Committee of Griffith University (MHIQ/11/20). All procedures conformed to the guidelines of the National Health and Medical Research Council.

Chikungunya disease patients - serum was obtained from chikungunya patients from twelve Brazilian cities (Aracaju, Itabaiana, Macambira, Campo do Brito, Nossa Sra do Socorro, Moita Bonita, Itabaianinha, Laranjeiras, Socorro, Malhador, Boquim and Capela) who were diagnosed at the University Hospital in Aracaju, Sergipe State, Brazil after the patients provided informed consent. The diagnosis of CHIKV infection was based on clinical symptoms, fever, skin exanthem, serological findings and a positive qRT-PCR test. Chronic serum samples were collected from patients with CHIKV infection confirmed by RT-qPCR and/or IgG/IgM testing and more than 90 days of symptoms, and acute serum samples were collected from patients with CHIKV RT-qPCR and/or IgM and in the first days of symptoms (mean time between disease onset and sample collection was 19.13 days and 258.46 days for the acute and chronic cohorts, respectively). The average age of the CHIKV patient cohort was 47.3 years old, and the distribution was 75% female, 25% male. The age is significantly higher in chronic patients compared to acute patients ($p < 0.01$). Control serum was obtained from matching healthy controls (tested negative for CHIKV RT-qPCR and/or IgG/IgM). The average age of the control group was 33.13 years old, and the distribution was 62.5% female, 37.5% male (see Supplementary Tables 1 and 2). The study was approved by the human ethics committee of the Federal University of Sergipe University Hospital, Brazil (reference number 1.486.302).

RRV disease patients - written informed consent was obtained from all patients as part of the Dubbo Infection Outcomes Study[70]. The study was approved by the Human Research Committee of the University of New South Wales (No. 04257). For this analysis, stored samples were available from RRV patients who reported symptom onset between days 16–22 (mean time 19.2 days) prior to sampling, and the serum and plasma samples tested positive for RRV-specific IgM. The average age cohort was 41 years old, and the distribution was 40% female, 60% male. Patients were asked to response to symptoms including fever, rash, malaise, lymphpad, headaches, arthralgia, myalgia activity, body pain, longer sleep, post-activity tired, poor sleep and tired activity using number codes 0–2 (0: none of the time or some of the time; 1: a good part of the time; 2 most of the time or all of the time). Serum from healthy individuals was provided by the Australian Red Cross with written and oral informed consent; this protocol was

approved by the Griffith University Human Research Ethics Committee (BDD/01/12/HREC). The average age for the control group was 46.2 years old, and the distribution was 50% female, 50% male (Supplementary Table 3). There is no significant difference between the age of the RRV-infected patients vs the control group.

## Cells and viruses

Cells were cultured at 37 °C in a 5% $CO_2$ atmosphere. Vero cells (ATCC CCL-81) were cultured in DMEM (Gibco, Thermo Fischer Scientific), supplemented with 10% fetal calf serum (FCS). These cells were determined to express FHL1 by western blot analysis. Human osteosarcoma (U2OS) cells (ATCC HTB-96) were maintained in DMEM supplemented with 10% FCS. Primary murine fibroblasts were obtained from the tail end of equal numbers of male and female WT or FHL1$^{-/-}$ mice. Briefly, mice were anesthetized with isoflurane. The tail end of 4 anesthetized mice were snipped and soaked in 80% ethanol (v/v) for ~1 min. The tail snips were then transferred into an Eppendorf tube containing 1 mL of collagenase Type III/DNase I in RPMI-1640 medium (Thermo Fisher) supplemented with 20% FCS and 1x anti-anti (Gibco, Thermo Fischer Scientific) and finely chopped. The tissues were then incubated overnight on a shaker at 37 °C. The next day, the tissues were pipetted up and down repeatedly to loosen all the cells and filtered through a 70 μm cell strainer. The filtered samples were then centrifuged at 200 × *g* for 5 min and the supernatant was discarded. The cell pellets were resuspended in 1 mL of RPMI-1640 medium supplemented with 20% FCS and 1× anti-anti and grown in a T25 cell culture flask. Once the cells reached 80% confluency, they were split into T75 flasks and maintained for experimental use.

RRV T48, CHIKV (LR2006-OPY1), MAYV and ONNV were rescued from respective infectious cDNA clones. The mutant virus CHIKV-ΔFHL1, nsP3 of which is unable to interact with FHL1 was generated on a background of CHIKV (LR2006-OPY1) by introduction of combination of 6 substitutions (F1762G, F1763G, T1780R, A1781S, Q1785G, and A1786T) previously shown to prevent its interaction with FHL1[30], into HVD of nsP3. The rescued viruses were amplified in Vero cells and the titers were determined by plaque assay on Vero cells.

## Plaque assay

Vero cells were seeded at a density of $2 \times 10^5$ cells per well in 12-well plates and incubated overnight at 37 °C in a humidified $CO_2$ incubator. The culture medium was aspirated, and the cells were washed with sterile PBS. The cells were inoculated with 200 μL of samples and incubated for 1 h at 37 °C in a humidified $CO_2$ incubator. An agarose overlay was added to each well and cultures were incubated for 48 h at 37 °C in a humidified $CO_2$ incubator. The cell monolayer was fixed with 1% formaldehyde and stained with 0.1% crystal violet. The virus titer was calculated as PFU per mL or tissue weight.

## Generation of FHL1 knockout mice

The *FHL1* gene is situated on chromosome Xq36 and encodes three different protein isoforms, FHL1A, FHL1B, and FHL1C[71]. The three isoforms of FHL1 were all knocked out, by two sgRNAs (5′ GTATCGCTGTCAAGTCAATA3′ and 5′CGTGGGTTGCGCATTATCTC3′) targeting 7572 bp which covered coding exons of FHL1. FHL1$^{-/-}$ knockout mice were made by injecting 20 ng/μL of Cas9 mRNA, 10 ng/μL of the sgRNAs, into the cytoplasm of fertilized one-cell stage embryos isolated from WT C57BL/6J breeders. Twenty-four hours later, two-cell stage embryos were transferred into the uteri of pseudopregnant mice. Viable offspring were genotyped by next-generation sequencing (Supplementary Fig. 1a). Out of the 18 mice born, four mice were validated via PCR to have the deleted allele (Supplementary Fig. 1b). The mice were genotyped; sequencing confirmed that all targeted alleles had the critical gene exons excised and should lead to a null allele (Supplementary Fig. 1c, d). These FHL1$^{-/-}$ mice were bred in-house to establish germline homozygosity (only deleted alleles present, no WT allele) (Supplementary Fig. 1e). Western blot analysis was carried out in the FHL1$^{-/-}$ mice bred in-house to further validate that these knockout mice were not expressing FHL1 (Supplementary Fig. 1f). These mice produced litters at normal frequency and with normal litter sizes, suggesting there is no defect in their fertility. No significant difference was observed in the populations of the major immune cells between the FHL1$^{-/-}$ and WT mice (Fig. 4b, Supplementary Fig. 3), indicating no developmental defects in the cell mediated immunity of these FHL1$^{-/-}$ mice.

## Mouse experiments

WT C57BL/6 mice were obtained from the Animal Resource Centre (ARC, Australia) whenever necessary. FHL1$^{-/-}$ mice were bred in-house. All mice were housed in Griffith animal house with food and water provided in the cage. The housing temperature was controlled at $21 \pm 2$ °C, with a humidity level of 50–60%. A 12 h light/12 h dark cycle was implemented.

Experiments involving chikungunya virus were carried out in a PC3/BSL3 facility, strictly adhering to rigorous safety protocols and guidelines. Experiments involving RRV, ONNV, and MAYV, on the other hand, were all conducted within a PC2/BSL2 facility, with the same level of commitment to stringent safety protocols and guidelines.

Equal numbers of 6–8 weeks old male and female WT C57BL/6 and FHL1$^{-/-}$ mice were infected by inoculation of $10^4$ PFU CHIKV, MAYV or ONNV diluted in PBS to a volume of 20 μL into the ventral/lateral side of the right foot; control mice were mock-infected with PBS. Mice were monitored daily for symptoms and weights; animals were sacrificed by $CO_2$ asphyxiation at experimental endpoints. CHIKV-, MAYV- or ONNV- induced footpad swelling was assessed by measuring the height and width of the perimetatarsal area of the ipsilateral hind foot using Kinchrome digital vernier calipers. Infection with RRV was performed as described above except animals were 20–22 days old and were inoculated subcutaneously in the thorax below the right forelimb. RRV infected mice were monitored daily, and clinical scores and weights were recorded until euthanasia. Clinical symptoms were scored as follows: 0, no disease signs; 1, ruffled fur; 2, mild hind limb weakness; 3, moderate hind limb weakness; 4, severe hind limb weakness and dragging of hind limbs; 5, complete loss of hind limb function; and 6, moribund. RRV-infected mice showing weight loss greater than 15% from the previous day with a clinical score of 6 were euthanized via asphyxiation.

## qRT-PCR

Total RNA extraction was carried out using TRIzol reagent (Life Technologies) following the manufacturer's instructions. One microgram of total RNA was reverse transcribed using random nonamer primers and MMLV reverse transcriptase (Sigma Aldrich, Inc, USA) according to the manufacturer's instructions.

qRT-PCR was carried out with 50 ng of cDNA template, Quanti-Tect primer assay kits (Qiagen, Hilden, Germany) and SYBR green real-time PCR reagent on a CFX96 Touch real-time system in a 96-well plate. The cycler conditions were as follows: (i) 95 °C for 15 min, 1 cycle and (ii) 3-step cycling: 94 °C for 15 s, 55 °C for 30 s and 72 °C for 30 s, 40 cycles. Data were normalized to HPRT housekeeping gene.

## Cytokine and chemokine profile analysis

Quadriceps muscles were harvested from CHIKV-infected WT and FHL1$^{-/-}$ mice at 3 and 7 dpi. Tissues were homogenized in T-PER (Thermo Fischer Scientific) with 1× HALT Protease inhibitor cocktail (Thermo Fischer Scientific) and cytokine and chemokine levels were analyzed using a Bio-Plex Pro Mouse Cytokine 23-plex Assay kit (Bio-Rad, #m60009rdpd) according to the manufacturer's instructions.

## Cell isolations for CyTOF and conventional fluorescence flow cytometry

**Quadriceps.** Mouse quadriceps were harvested into 2 mL round-bottom tubes containing 1 mL of collagenase digestion solution at 3 and 7 dpi. The tissues were minced into slurries with sterile scissors. The slurries were transferred into 12-well plates (one well per sample) containing 2 mL of digestion solution per well and incubated at 37 °C for 1 h. The slurries were resuspended with a 1 mL plastic Pasteur pipette in RPMI-1640 medium and passed through a prewetted 70-μm cell strainer into new 10 mL tubes. The samples were centrifuged at 400 × g for 5 min at 4 °C. The supernatants were decanted, and the samples were resuspended in FACS buffer (5 g bovine serum albumin in 500 mL of PBS and 2 mL of 0.5 M EDTA). The samples were passed through a prewetted 30 μm cell strainer into new 10 mL tubes and centrifuged at 400 × g for 5 min at 4 °C. The supernatants were discarded, and the samples were resuspended in 2 mL of FACS buffer and kept on ice until staining.

### Spleen

Mouse spleens were harvested into 2 mL round-bottom tubes containing 1 mL of RPMI-1640 medium at 3 and 7 dpi. The samples were placed onto a prewetted 70-μm cell strainer and ground with a syringe plunger into a new 10 mL tube. The samples were centrifuged at 400 × g for 5 min at 4 °C. The supernatants were discarded, and the samples were resuspended gently in the remaining RPMI-1640 medium. Miltenyi Lysing Solution was added to each sample, vortexed briefly and left to incubate at room temperature for 2 min. The samples were then centrifuged at 400 × g at room temperature for 5 min. The supernatants were decanted, and the samples were resuspended in 2 mL of FACS buffer and kept on ice until staining.

**Lymph nodes.** Mouse lymph nodes were harvested into 2 mL round-bottom tubes containing 1 mL of RPMI-1640 medium at 3 and 7 dpi. The samples were filtered through a prewetted 70-μm cell strainer and ground with a syringe plunger into a new 10 mL tube. The samples were centrifuged at 400 × g for 5 min at 4 °C. The supernatants were decanted, and the cells were resuspended in 2 mL of FACS buffer and kept on ice until staining.

The gating strategy shown in Supplementary Information Gating Strategy.

### Cell staining for CyTOF

Isolated mouse splenocytes and popliteal draining lymph node cells were counted and viability determined by trypan blue exclusion. One million cells per sample were barcoded with anti-CD45 antibodies conjugated to various metal tags and incubated in FACS buffer containing 5 μg/mL anti-CD16/32 Fc block for 30 min at 4 °C, before they were washed with 2 mL FACS buffer and pelleted at 400 × g for 5 min at 4 °C. CD45-barcoded samples were resuspended in 1 mL FACS buffer and combined for subsequent labelling. The cells were then stained with 200 μL of 5 μM cisplatin solution in PBS for 5 min at room temperature, before quenching with 2 mL of FACS buffer and centrifuging at 400× g for 5 min at 4 °C. The supernatant was discarded and cells were labelled with metal-tagged antibodies (Supplementary Table 4) against surface markers in FACS buffer and incubated for 30 min at 4 °C, according to previously published protocols[72]. The cells were then washed with 3 mL FACS buffer and centrifuged at 400 × g for 5 min at 4 °C and supernatant discarded. The cells were then incubated with secondary metal-tagged antibodies against the appropriate fluorophores in FACS buffer for 30 min at 4 °C before they were washed with 3 mL FACS buffer and centrifuged at 400 × g for 5 min at 4 °C. For intracellular staining, the cells were fixed, permeabilized and washed with eBioscience Foxp3/Transcription Factor Staining Buffer Set according to manufacturer's protocols, followed by incubation with metal-tagged intracellular antibodies for 30 min at 4 °C. The cells

were then washed with wash buffer and pelleted at 900 × g for 10 min and subsequently stained with 500 μL Iridium DNA intercalator in 4% paraformaldehyde for 20 min at room temperature. Finally, the cell samples were washed in FACS buffer, followed by ultrapure water and resuspended in Maxpar Cell Acquisition Solution (Fluidigm) before sample acquisition with the CyTOF Helios Mass Cytometer.

Analysis of acquired mass cytometry data was undertaken using FlowJo v10.8 with the UMAP plugin v3.1[73]. Sample data was imported into FlowJo, cells identified by labelled DNA content, doublets and dead cells removed, and individual samples were de-multiplexed and identified according to CD45 barcode channels. Samples from each tissue type were then concatenated and computational analysis was performed using UMAP in FlowJo for dimensionality reduction and cell phenotype visualization. Cell type clusters were identified on the UMAP output using traditional immune cell markers. Statistical analyses of the fold-change in expression of various markers were performed in GraphPad Prism v9.4.

### Cell staining for conventional fluorescence flow cytometry

Cells were resuspended in staining buffer (PBS with 2% FCS; 5 mM EDTA), blocked with anti-CD16/32 Fc block (BD, Biosciences, Franklin Lakes, NJ, USA), and labeled with fluorochrome-conjugated antibodies for 1 h. Antibodies against CD45 (HI30; BD, Biosciences), CD3 (145-2C11; Invitrogen), CD4 (RM4-5; BioLegend), CD8 (53-6.7; BioLegend), Ly6C (HK1.4; BioLegend), Ly6G (1A8; BD, Bioscience), CD11b (M1/70; BioLegend) and NK1.1 (PK136; BD, Bioscience) were used, and a near infrared (NIR) LIVE/DEAD stain (Life Technology) was used to exclude dead cells. Counting beads (Spherobeads, BD) were added to samples before acquisition. Samples were acquired on a BD LSRFortessa flow cytometer, and data analysis was performed using FlowJo 10.7.

### ELISA

ELISA analysis of human serum samples was carried out using the Human FHL1 ELISA Kit (Assay Genie, HUDL01076) following the manufacturer's instructions. For mouse samples, ELISA analyses were carried out as follows. Maxisorp ELISA plates were coated with 2 μg/mL anti-FHL1 monoclonal antibodies (mAbs) (Genscript) overnight in carbonate buffer (3.39 g of sodium carbonate and 5.7 g of sodium bicarbonate in 1 L of ddH$_2$O) at 4 °C. The carbonate buffers and samples were removed, and the plates were washed three times with PBS and then blocked for 1 h at 37 °C with 5% skim milk in PBS. Then, the plates were washed with PBS three times, and the indicated mouse samples were added to the wells and incubated at 37 °C for 1 h. The plates were washed 3 times with wash buffer (0.05% Tween 20 (v/v) in PBS-D) and then incubated with anti-mouse IgG HRP (Sigma–Aldrich) at a dilution of 1:7500 for 1 h at 37 °C. The plates were washed 3 times with wash buffer, TMB substrate (Thermo Fisher) was added, and the plates were incubated for 15 min in the dark at room temperature. H$_2$SO$_4$ was added to the plates to a final concentration of 0.1 M, and the absorbance was read at 450 nm using a POLARstar Omega plate reader.

### Western blot

Protein was extracted using radioimmunoprecipitation (RIPA) assay buffer with protease inhibitor cocktail and subsequently measured using a biospectrometer (Eppendorf). Protein (20 μg) was added to each lane of Bolt 8%, Bis–Tris, 1.0 mm, Mini Protein Gels (Thermo fisher) and was then run for 45 min (constant mode, 180V). Subsequently, the gels were transferred to polyvinylidene membranes using iBlot 2 Gel transfer device (Thermo fisher). Further, blocking was done with 5% skim milk-0.2% Tween 20-PBS solution for 1 h and incubated with anti-beta actin (1:1000, Abcam) antibody and anti-FHL1 (1:1000, Invitrogen, MA5-25135) antibody for 2 h, for each membrane, at room temperature. Next, they were washed three times and incubated with anti-mouse HRP antibody (1:3000, Cell Signaling Technology, 7076S)

for 1 h at room temperature and imaged after adding Immobilon Western Chemiluminescent HRP substrate in a ChemiDoc MP gel imaging system (BioRad).

## Immunofluorescence staining and confocal microscopy
Mice were euthanized and perfused with PBS. The quadricep muscles were harvested, transferred into Falcon tubes and fixed with 4% PFA on a rotator at 4 °C for 12 h. Then the tissues were immersed in 30% (v/v) sucrose for 24 h at 4 °C, embedded in OCT compound (Sakura Finetek) and frozen overnight at −80 °C. The OCT blocks were sectioned to 20 μm, mounted on a frost slide and permeabilized with ice-cold acetone. Sections were then blocked with DAKO-protein block and immunolabeled with anti-FHL1 (Genscript) and nuclei were counterstained with DAPI (Thermo Fischer Scientific). Slides were mounted with ProLong Gold antifade agent (Thermo Fisher). Images were acquired using an Olympus FV3000 confocal microscope with a 20× objective and processed using FV31S-DT (Olympus Software).

## Histological staining
Mice were euthanized and perfused with PBS. The quadricep muscle and joint tissues were harvested, transferred into 15 mL Falcon tubes and fixed with 4% PFA on a rotator at 4 °C overnight. The quadriceps muscle tissues were washed with PBS for 1 h the next day and stored in 70% ethanol for paraffin infiltration. The ankle tissues were decalcified in 14% EDTA for 10 days (solution changed every 2 days) after 48 h of PFA fixation and stored in 70% ethanol for paraffin infiltration. Tissues were infiltrated with paraffin using a Leica TP1020 tissue processor and embedded into paraffin blocks using a Myr Tissue Embedding Centre EC 500. The paraffin blocks were sectioned into 5-μm-thick sections using a Leica RM2245 Semi-Automated Rotary Microtome, and sections were transferred onto a water bath set at 37 °C to unfurl any wrinkled sections. The unfurled sections were collected using positively charged slides. The slides were dried overnight before being subjected to deparaffinization and rehydration. The rehydrated slides were then stained with H&E or safranin-O according to the manufacturer's protocol. The stained slides were imaged with a Leica Aperio AT2 and exported to QuPath analysis software for quantification of cellular infiltrates and analysis of tissue damage.

## Viral growth kinetics assay
Primary murine fibroblasts generated from WT or FHL1$^{-/-}$ mice (equal sex distribution) were cultured to at least passage 3 and maintained in RPMI-1640 medium supplemented with 20% FCS and anti-anti. Cells were seeded in 12-well plates; infected with CHIKV, RRV, MAYV or ONNV at a multiplicity of infection (MOI) of 0.01 for 1 h, after which inoculum was removed, cells washed once with PBS and then maintained in RPMI-1640 medium supplemented with 2% FCS. Supernatants were collected at indicated the time-points after infection and tittered on Vero cells by plaque assay.

## PRNT assay
Serum was collected from mice vaccinated with CHIKV-ΔFHL1 at 14 days post-vaccination. Serum samples were heat inactivated at 56 °C for 30 min, prior to twofold serial dilutions of 10-, 20-, 40-, 80-, 160-, 320-, 640- and 1280-folds. The diluted serum was combined with 1000 PFU/ml of CHIKV, RRV, MAYV or ONNV at a 1 to 1 ratio, and incubated at 37 °C for 1 h. The plaque forming units in these serum-virus mixtures were enumerated by plaque assay. The levels of the neutralizing antibody in the serum samples were expressed as the reciprocal of the fold-dilution which reduced 50% of the original virus titer (PRNT$_{50}$).

## Pull-down assay
The CMV-EGFP-HVD plasmid used for the expression of EGFP fused with the HVD in CHIKV nsP3 has been previously described[74].

Sequences encoding for HVD of nsP3 in CHIKV-ΔFHL1, ONNV, ONNV-ΔFHL1 (harbouring the same set of mutations as CHIKV-ΔFHL1), RRV and MAYV were inserted into the same expression vector. 15 μg of each of obtained expression plasmids were used to transfect approximately $5 \times 10^6$ U2OS cells grown on 100-mm cell culture dishes using Lipofectamine LTX with PLUS reagent (Thermo Fisher Scientific). At 24 h post-transfection, the cells were collected and washed with PBS. 10% of collected cells were suspended in PBS, equal volume of 2× Laemmli sample buffer was added and proteins were denatured by boiling for 5 min. Remaining cells were lysed using 700 μL of cold lysis buffer (20 mM HEPES, pH 7.2; 150 mM NaCl; 2 mM MgCl$_2$; 100 mM K-acetate; 1% Triton X-100; 0.1% Tween-20; and 1 tablet of Pierce protease inhibitor (Thermo Fisher Scientific) per 50 mL of lysis buffer). After lysing on ice for 30 min, cell debris was removed by centrifugation at 15,000 × $g$ for 10 min at 4 °C. The supernatant was transferred to a clean tube and incubated with EGFP-binding magnetic beads (GFP-trap M, ChromoTek) for 1 h at 4 °C. After incubation, the beads were washed four times with lysis buffer, and bound proteins were denatured by boiling in Laemmli sample buffer for 5 min. Obtained samples were subjected to SDS–PAGE in 10% gels. The separated proteins were transferred to polyvinylidene difluoride membranes and visualized using antibodies against EGFP (in-house), G3BP1 (Santa Cruz Biotechnology, sc-365338) and FHL1 (Proteintech, 10991-1-AP). An antibody against β-actin (Santa Cruz Biotechnology, sc-47778) was used to detect the loading control. Membranes were washed 3 times with PBS, incubated with the corresponding secondary antibodies conjugated to fluorescent labels (Li-Cor), washed and imaged using a Li-Cor Odyssey Fc imaging system.

## Statistical analysis
All required statistical approaches were applied for data comparison whenever needed using GraphPad Prism, version 9. ELISA for RRV disease patients' serum and plasma samples, ELISA for mouse serum samples, qRT-PCR of mouse specimens, viral titer analyses of mouse specimens, histological analyses and flow cytometry analyses were statistically analyzed using Mann–Whitney test. ELISA for chikungunya patients' serum samples were statistically analyzed by one-way ANOVA with a Kruskal–Wallis posttest. Mice disease and weight were statistically analyzed by two-way ANOVA with a post hoc Bonferroni test. Multiplex protein analysis of CHIKV-infected quadriceps homogenate were statistically analyzed by two-way ANOVA with a post hoc Holm–Sidak test. Mass cytometry data were analyzed using two-way ANOVA with a post hoc Tukey test. Equal numbers of male and female mice were allocated to each group and weighed to avoid body-mass dependent bias. Scoring of disease signs following viral infection was carried out by two researchers. All data are presented as the mean ± SEM unless otherwise stated.

## Reporting summary
Further information on research design is available in the Nature Portfolio Reporting Summary linked to this article.

# Data availability
Datasets generated and/or analysed during the study are included in the main manuscript or appended as supplementary data. Source data are provided with this paper.

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

## Acknowledgements

The generation of the FHL1$^{-/-}$ mice used in this study was supported by Phenomics Australia, the Australian Government through the National Collaborative Research Infrastructure Strategy (NCRIS) program and the Australian National Health and Medical Research Council (NHMRC) Grant (APP1106411 to S.M). We thank Professor Christina Mitchell (Monash University) for kindly providing us with the primary FHL1 antibody used in this study. Supplementary Fig. S1a was generated using BioRender. This study was supported by a grant from the Australian NHMRC to S.M. (APP1184879). S.M. is the recipient of an NHMRC Senior Research (APP1154347). W.H.N. is a recipient of a Griffith University Postgraduate Scholarship. R.P.A. is supported by a FINEP Grant (0116005600). A.M. is the recipient of program grant (PRG1154) from Estonian Research Council.

## Author contributions

S.M. conceived and supervised this study. S.M., W.H.N., X.L., A.R.L., M.M.T., A.M., R.P.A., and N.J.C.K. designed experiments. W.H.N., X.L., Z.L.L., C.N.O.S., L.S.M., S.K., S.W., and A.J.K. performed experiments. W.H.N., X.L., Z.L.L., C.N.O.S., L.S.M., A.T., J.R.F., and N.J.C.K. analyzed data. M.J.H., A.R.L., A.M., and R.P.A. provided essential reagents. S.M., W.H.N., and X.L. wrote the manuscript. All authors reviewed and edited the manuscript.

## Competing interests

The authors declare no competing interests.

## Additional information

[1]Emerging Viruses, Inflammation and Therapeutics Group, Menzies Health Institute Queensland, Griffith University, Gold Coast, QLD 4222, Australia. [2]Global Virus Network (GVN) Centre of Excellence in Arboviruses, Griffith University, Gold Coast, QLD, Australia. [3]School of Pharmacy and Medical Sciences, Griffith University, Gold Coast, QLD, Australia. [4]Viral Immunopathology Laboratory, Infection, Immunity and Inflammation Research Theme, Charles Perkins Centre, School of Medical Sciences, Faculty of Medicine and Health, The University of Sydney, Sydney, NSW 2006, Australia. [5]Sydney Institute for Infectious Diseases, Sydney Medical School, The University of Sydney, Sydney, NSW 2006, Australia. [6]Division of Immunology and Molecular Biology Laboratory, University Hospital/EBSERH, Federal University of Sergipe (UFS), Aracaju, Brazil. [7]The Walter and Eliza Hall Institute of Medical Research, Melbourne, VIC, Australia. [8]Department of Medical Biology, The University of Melbourne, Parkville, VIC 3050, Australia. [9]Institute of Technology, University of Tartu, Tartu, Estonia. [10]Viral Immunology Systems Program, Kirby Institute, University of New South Wales, Kensington, NSW, Australia. [11]Instituto de Ciencias Biologicas, Universidade Federal de Minas Gerais, Belo Horizonte, Minas Gerais, Brazil. [12]These authors contributed equally: Wern Hann Ng, Xiang Liu.
✉e-mail: s.mahalingam@griffith.edu.au

