## [Peer Review File · Nature Communications]

Reviewers' Comments:

Reviewer #1:

Remarks to the Author:

This is a resubmission of manuscript entitled "FHL1 promotes chikungunya and o'nyong-nyong virus infection and pathogenesis – implications for arthritogenic alphavirus vaccine design". The authors have performed a nice set of revisions on this manuscript and have addressed nearly all of the concerning points raised by the reviewers. Similar to what has been published previously by Mertens et al. this group finds that FHL1A is a critical host factor in CHIKV pathogenesis. They extensively characterize a newly generated FHL1A ko mouse line in the context of the adolescent arthritis model and also demonstrate its potential role as a vaccine candidate. In response to the reviewers' comments they have added additional characterization of this novel mouse line which greatly enhances the manuscript and they have compared its efficacy as a vaccine candidate against a CHIKV-3del5 strain which is also being developed as a vaccine candidate. While concerns were raised about the novelty of these findings, the new data which evaluates FHL1 binding to nsP3 of different alphavirus strains and finds that it only binds to CHIKV and ONNV but not RRV and MAYV is quite striking, and given that there is no phenotype in the RRV and MAYV infection in FHL1 ko mice provides important new data to the field. Overall, I think this is now a well written and important study that contributes to our understanding of CHIKV pathogenesis.

Reviewer #3:

Remarks to the Author:

The authors have addressed all of my concerns and have significantly modified the manuscript. I have no further concerns.

Reviewer #4:

Remarks to the Author:

The authors have done a nice job responding to the reviewer's comments which have strengthened their paper. However, there are still major concerns that call into question the rigor and conclusions of the studies. I suggest these be addressed as they are critical for the authors points and for the idea of novelty.

1. The novelty of the work is still at question. I agree with many of the points the authors brought up to justify why their work is novel and I like the idea of the deltaFHL1 virus as a vaccine. However, there is one major concern that questions the conclusions of the vaccine portion of the paper.

The authors make mutations in the HVD fusion proteins that nicely ablate FHL1 binding. They transfer these mutations to a CHIKV-deltaFHL1 virus which is attenuated in Vero cells. However, Lukask et al., have shown that Vero E6 cells don't express FHL1 and that a virus containing similar mutations replicates well in Vero E6 cells. Given that Vero E6 cells don't express FHL1, it suggests that your mutant may be attenuated for another reason that may completely unrelated to FHL1, making conclusions that disrupting FHL1:HVD is the true mechanism for the vaccine hard to justify. The fact that the delta FHL1 virus behaves exactly like the 3del5 makes me think it could general replication.

One explanation here is that the Vero cells you are using do have FHL1 and a simple western blot would answer that. If the Vero cells don't have FHL1, then it is very hard to pin the mechanism on

FHL1-binding opposed to just a defect in RNA replication of another reason. In this case, the conclusion of blocking FHL1 binding as a vaccine won't stand up. Another explanation could be the virus strain used. Either way, these points are critical to address.

2. The authors don't use litter mate controls for any mouse experiments. I found this strange given they bred their FHL1 knockout mice in-house. This is particularly concerning with immunology (and virology) studies as we know mice from different sources can yield different results, even in viral titers. It would have been better to generate isogenic WT and FHL1 KO mice or at least breed WT and FHL1 KO mice in the same facility to control for microbiome, etc. I'm not sure the results will change, but taking these things into consideration is important. Without repeating experiments, the authors should justify why they didn't use litter mate controls and add this as a limitation to their study in the discussion.

Response to Reviewers

We thank the reviewers for their input, and we believe their comments have greatly improved our manuscript. We have now incorporated additional modifications and results have been included in this new version of the manuscript as summarized below:

1) Confocal microscopy

- We have generated and updated more clearer confocal microscopy images to validate the absence of FHL1 expression in GKO mice (See **Fig. 1b**) and for the identification of FHL1-expressing cell types (See **Extended Data Fig. 5**).

2) Western blot

- We have generated Western Blot data to provide further validation of the FHL1 knockout mice model. This data is now included in **Extended Data Fig 1f**.
- We have also shown that the Vero cell lines that we used (Vero CCL-81) express FHL1 using Western Blot. Please see **Figure R4.1** for details.

Point-by-point Responses:

Reviewer #1 (Remarks to the Author):

This is a resubmission of manuscript entitled “FHL1 promotes chikungunya and o’nyong-nyong virus infection and pathogenesis – implications for arthritogenic alphavirus vaccine design“. The authors have performed a nice set of revisions on this manuscript and have addressed nearly all of the concerning points raised by the reviewers. Similar to what has been published previously by Mertens et al. this group finds that FHL1A is a critical host factor in CHIKV pathogenesis. They extensively characterize a newly generated FHL1A ko mouse line in the context of the adolescent arthritis model and also demonstrate its potential role as a vaccine candidate. In response to the reviewers' comments they have added additional characterization of this novel mouse line which greatly enhances the manuscript and they have compared its efficacy as a vaccine candidate against a CHIKV-3del5 strain which is also being developed as a vaccine candidate. While concerns were raised about the novelty of these findings, the new data which evaluates FHL1 binding to nsP3 of different alphavirus strains and finds that it only binds to CHIKV and ONNV but not RRV and MAYV is quite striking, and given that there is no phenotype in the RRV and MAYV infection in FHL1 ko mice provides important new data to the field. Overall, I think this is now a well written and important study that contributes to our understanding of CHIKV pathogenesis.

We appreciate the reviewer's review of our manuscript and the recognition of our revisions. The feedback has been invaluable in improving the quality and significance of our study. We are delighted to hear that our efforts in addressing the concerns raised have contributed to making the manuscript more impactful.

Reviewer #3 (Remarks to the Author):

The authors have addressed all of my concerns and have significantly modified the manuscript. I have no further concerns.

We thank the reviewer for the review of our revised manuscript. We believe that the implementation of the comments made previously has greatly improved our manuscript.

Reviewer #4 (Remarks to the Author):

The authors have done a nice job responding to the reviewer's comments which have strengthened their paper. However, there are still major concerns that call into question the rigor and conclusions of the studies. I suggest these be addressed as they are critical for the authors points and for the idea of novelty.

(R4) Point 1: *The novelty of the work is still at question. I agree with many of the points the authors brought up to justify why their work is novel and I like the idea of the deltaFHL1 virus as a vaccine. However, there is one major concern that questions the conclusions of the vaccine portion of the paper.*

The authors make mutations in the HVD fusion proteins that nicely ablate FHL1 binding. They transfer these mutations to a CHIKV-deltaFHL1 virus which is attenuated in Vero cells. However, Lukask et al., have shown that Vero E6 cells don't express FHL1 and that a virus containing similar mutations replicates well in Vero E6 cells. Given that Vero E6 cells don't express FHL1, it suggests that your mutant may be attenuated for another reason that may completely unrelated to FHL1, making conclusions that disrupting FHL1:HVD is the true mechanism for the vaccine hard to justify. The fact that the delta FHL1 virus behaves exactly like the 3del5 makes me think it could general replication.

One explanation here is that the Vero cells you are using do have FHL1 and a simple western blot would answer that. If the Vero cells don't have FHL1, then it is very hard to pin the mechanism on FHL1-binding opposed to just a defect in RNA replication of another reason. In this case, the conclusion of blocking FHL1 binding as a vaccine won't stand up. Another explanation could be the virus strain used. Either way, these points are critical to address.

We thank the reviewer for providing insightful feedback, and we recognize the importance of addressing the concern related to the expression of FHL1 in Vero cells. We wish to clarify that our study utilized Vero CCL-81 cells rather than Vero E6 cells (**see line 518**). To validate the presence of FHL1 in Vero CCL-81 cells, we have carried out Western Blot analyses and have shown that Vero CCL-81 cells do express

FHL1 (**Figure R4.1**). We have included a statement saying that Vero CCL-81 expresses FHL1 in **line 520**.

Figure R4.1. Western Blot data loaded with heart samples from FHL1^{-/-} mice, WT mice and Vero CCL-81. Mouse samples and Vero cells were harvested and lysed with RIPA buffer containing protease inhibitors. Samples were then processed for Western Blot, loaded into Bolt™ Bis-Tris Plus Mini Protein Gels, 10%, and ran under 180V for 45 mins. Gels were then transferred onto a membrane using iBlot2. The membranes were incubated in blocking buffer for 1 hour with gentle agitation on an orbital shaker before incubation with primary antibody for 1 hour at room temperature. The membranes were then washed thrice with TBST (15 mins each wash) and incubated with secondary antibody for 45 mins at room temperature. The membranes were washed again with TBST thrice (15 mins each wash) and were then subjected to electrochemiluminescence development. Images were obtained using ChemiDoc MP gel imaging system and processed with Adobe Illustrator.

Moreover, we would like to reference a previous study (DOI:<https://doi.org/10.3389/fgene.2022.801382>) that underscores a substantial genetic disparity between Vero CCL-81 and Vero E6 cell lines. Notably, Vero E6 cells have been reported to exhibit monosomy for the X chromosome. Given that FHL1 is located on the X chromosome, this genetic distinction could potentially explain the discrepancy in FHL1 expression between Vero CCL-81 and Vero E6 cell lines. Consequently, the FHL1 expression observed in Vero CCL-81 cells aligns with our experimental findings.

(R4) Point 2: *The authors don't use litter mate controls for any mouse experiments. I found this strange given they bred their FHL1 knockout mice in-house. This is particularly concerning with immunology (and virology) studies as we know mice from different sources can yield different results, even in viral titers. It would have been better to generate isogenic WT and FHL1 KO mice or at least breed WT and FHL1 KO mice in the same facility to control for microbiome, etc. I'm not sure the results will change, but taking these things into consideration is important. Without repeating experiments, the authors should justify why they didn't use litter mate controls and add this as a limitation to their study in the discussion.*

We appreciate the reviewer's consideration and insight regarding the choice of controls in our mouse experiments. We acknowledge that not employing littermate controls is a limitation in our study and have now included a discussion in the manuscript that addresses the absence of littermate controls as a limitation of our study in line 455-469.

Briefly, it is important to point out that FHL1^{-/-} mice are not commercially available and as such, had to be bred in-house after generation of these mice. To minimize the chances of genetic drift, we crossbred FHL1^{-/-} mice with WT counterparts sourced from the Australian Animal Resources Centre (ARC). We were unable to generate littermate controls in our mouse experiments due to constraints in the breeding capacity of our animal facility and as such, had to source WT mice from the Australia Animal Resources Centre (ARC). The FHL1^{-/-} strain was established using WT mice (C57Bl/6 background) sourced from ARC. These WT mice were also used as the control group for our study. To further ensure the validity and reliability of our data, we have also carried out flow cytometry analysis in the PBS groups and have shown that there were no differences between the major immune cells observed between the FHL1^{-/-} and WT mice (Fig. 4b, Extended Data Fig. 3). These measures collectively contribute to the validity and reliability of the conclusions drawn from this study.

Reviewers' Comments:

Reviewer #4:

Remarks to the Author:

The authors have nicely addressed my concerns. Great work!

Response to Reviewer #4

Reviewer 4 (R4)

The authors have nicely addressed my concerns. Great work!

Response: We thank the reviewer for the comments.